

# Characterising recent drought events in the context of dry-season trends using state-of-the-art reanalysis and remote-sensing soil moisture products

Martin Hirschi[1], Bas Crezee[1,3], Pietro Stradiotti[2], Wouter Dorigo[2], Sonia I. Seneviratne[1]

[1]ETH Zurich, Institute for Atmospheric and Climate Science, Universitätstrasse 16, 8092 Zürich, Switzerland
[2]TU Wien, Department of Geodesy and Geoinformation, Wiedner Hauptstrasse 8-10, 1040 Vienna, Austria
[3]now at: Federal Office of Meteorology and Climatology MeteoSwiss, 8058 Zürich-Airport, Switzerland

*Correspondence to*: Martin Hirschi (martin.hirschi@env.ethz.ch)

**Abstract.** Drought events have multiple adverse impacts on environment, society, and economy. It is thus crucial to monitor and characterise such events. Here, we compare the ability of selected state-of-the-art long-term reanalysis and remote-sensing products to represent major seasonal and multi-year drought events in the 2000–2020 period globally. We focus on soil moisture (or agroecological) drought and place the results in the context of trends in dry-season soil moisture. We consider surface and root-zone soil moisture from ERA5, the related ERA5-Land, and MERRA-2 reanalysis products, the ESA CCI remote-sensing surface soil moisture products (encompassing an ACTIVE, a PASSIVE and a COMBINED product), as well as its near real-time counterpart produced within C3S. In addition, we use a new root-zone soil moisture dataset derived from the ESA CCI COMBINED product. Except for ESA CCI surface and root-zone soil moisture, the considered products offer opportunities for drought monitoring since they are available in near real-time.

We analyse 18 documented drought events within predefined spatial and temporal bounds derived from scientific literature. Based on standardised daily anomalies of surface and root-zone soil moisture, the drought events are characterised by their severity (the time accumulated standardised anomalies), magnitude (the minimum of the standardised anomalies over time), duration, and spatial extent. Product deviations in drought severity and magnitude are then placed in the context of trends in dry-season soil moisture, and potential reasons for diverging global soil moisture trends in the products are further investigated.

All investigated products capture the considered drought events. Overall, responses of surface soil moisture tend to be weakest for the ACTIVE remote-sensing products in all metrics, but most pronounced in the drought magnitudes. Also, MERRA-2 shows lower magnitudes than the other products. Except for the COMBINED products, the remote-sensing products tend to underestimate the spatial extents of larger droughts. Product differences in drought severity and magnitude for single events are consistent with the differences in dry-season soil moisture trends. These trends are globally diverse and



partly contradictory between products. ERA5, ERA5-Land and the COMBINED products show larger fractions of drying trends, MERRA-2 and the C3S ACTIVE and PASSIVE products more widespread wetting trends. MERRA-2 surface air temperature shows regionally negative biases in trends compared to a ground observational product, which suggests that this reanalysis product underestimates drought trends. Also, the comparison with trends in selected land-surface characteristics and bioclimatic indicators shows that dry-season soil moisture trends may be affected by retrieval or modelling artifacts in some cases.

In the root zone (based on the reanalysis products and the ESA CCI root-zone soil moisture dataset), the droughts are dampened in magnitude and smaller in spatial extent but show a tendency to prolonged durations. Based on the overall observational evidence and the consideration of the respective limitations of the included products, the present analyses suggest a consistent tendency towards drying during the last two decades in some regions, namely in parts of central Europe, in a region north of the Black Sea/Caspian Sea, in southern Africa, and in parts of Australia, Siberia and South America.

## 1 Introduction

Droughts have multiple impacts on environment, society, and economy, including substantial impacts on agriculture, ecosystems, and public water supply (Stahl et al., 2016; Seneviratne et al., 2021). Furthermore, droughts can act as trigger for other natural hazards at the sub-continental scale, including increased wildfire activity (Gudmundsson et al., 2014). Through feedback with the atmosphere, the prevailing dry conditions may further enhance air temperatures and trigger heat extremes (e.g., Miralles et al., 2014; Hirschi et al., 2011; Mueller and Seneviratne, 2012).

The recent IPCC AR6 report assessed three types of droughts (Seneviratne et al., 2021; Douville et al., 2021): meteorological droughts based on precipitation deficits, agricultural and ecological droughts – here referred to as "agroecological droughts" (Zaitchik et al., 2023) – related to deficits in soil moisture and other measures of changes in the land water balance, and hydrological droughts related to streamflow deficits. The most impact-relevant drought types are agroecological and hydrological droughts. The IPCC AR6 report concluded that a number of regions of the world are affected by increases in agroecological droughts (Seneviratne et al., 2021), mostly due to increases in evapotranspiration (Padron et al., 2020). Thus, monitoring and characterising soil moisture droughts is crucial, and will become more important with ongoing global warming.

The primary driver of agroecological and hydrological droughts is a lack of precipitation (meteorological drought; see e.g., Seneviratne, 2012; Seneviratne et al., 2021; Liu et al., 2020). Increased evapotranspiration due to enhanced radiation, wind speed, or vapor pressure deficit (itself linked to temperature and relative humidity) can further intensify the water shortage and lead to critical soil moisture values (agroecological drought) inducing e.g., adverse impacts on vegetation development



(due to increased water stress) and crop yield reduction/failure (e.g.,Teuling et al., 2013; Seneviratne et al., 2021; Bueechi et
al., 2023). Furthermore, pre-conditioning (pre-event soil moisture, surface, snow and/or groundwater storage) can contribute
to the emergence of agroecological and hydrological droughts (Koster et al., 2010). Under strong droughts, soil moisture can
also become limiting for evapotranspiration, thus reducing the evaporative cooling effect (e.g., Miralles et al., 2014;
Seneviratne et al., 2010).

Based on varying data products, regional soil moisture drying trends have been reported, e.g., for East Asia (e.g., Jia et al.,
2018; Cheng et al., 2015), Western and Central Europe (e.g., Trnka et al., 2015; Scherrer et al., 2022), and the Mediterranean
(e.g., Hanel et al., 2018; Moravec et al., 2019). Also, global studies have documented soil moisture drying during past
decades for several regions (Albergel et al., 2013; Gu et al., 2019; Preimesberger et al., 2021; Dorigo et al., 2012). However,
the involved products show partly considerable differences in the global patterns and magnitudes of the soil moisture drying.

Focussing on agroecological drought, here we systematically characterise documented major seasonal and multi-year
drought events in the 2000–2020 period globally and test the ability of selected state-of-the-art reanalysis (ERA5, the offline
ERA5-Land, and MERRA-2) and long-term remote-sensing products (ESA CCI soil moisture, and its near real-time
counterpart produced within the Copernicus Climate Change Service, C3S) to capture these events. In addition, the 2022
drought in Europe is used to introduce the methodology for characterising the drought events, based on near real-time
products only. The drought events are selected based on scientific literature and drought reports, providing the temporal and
spatial bounds for the analysis. We analyse product differences in the representation of the droughts by placing the results in
the context of product-specific global dry-season soil moisture trends. Also, drivers and controls of the dry-season soil
moisture trends are analysed by comparing those to trends in relevant variables of the land water balance, bioclimatic
indicators and land-surface characteristics that potentially affect the stability of the soil moisture retrieval. Since in situ
observations of soil moisture are still scarce and not continuously available in space and time over long time periods (Dorigo
et al., 2011; Dorigo et al., 2021c), reanalysis and merged remote-sensing products provide an alternative for global long-term
timeseries to investigate soil moisture droughts and related drying trends on supra-regional scales. Given the lack of widely
available ground data of soil moisture, we rely on well documented drought events and focus on the relative behaviour of the
products within the temporal and spatial bounds of the events. Thus, we do not aim for a in situ validation of the products
regarding their representation of the considered drought events but focus instead on the product ensemble to identify the
products with larger deviations from the majority and collect convergence of evidence. The considered products, in
particular the ones from the Copernicus Climate Data Store (CDS), also offer new opportunities for monitoring of ongoing
droughts and applications like drought index insurances (Vroege et al., 2021), since they are available in near real-time.



## 2 Data

### 2.1 Remote sensing and reanalysis soil moisture

For the drought characterization, soil moisture from both the near-surface soil layer as well as the root zone is considered. Despite the overall strong correlation of surface soil moisture with deeper soil layers, evapotranspiration and vegetation processes might be more sensitive to variations of root-zone soil moisture, in particular under very dry conditions (Hirschi et al., 2014). The surface layer corresponds to roughly 0–5 cm depth (according to GCOS, 2016) and covers the penetration depth of microwave remote sensing soil moisture products. Note that this upper soil layer depth may slightly vary per product depending on the microwave sensing frequency or the land-surface model. For the root zone, the soil layer of 0–100 cm depth is considered.

### 2.1.1 ESA CCI soil moisture

The European Space Agency (ESA) Climate Change Initiative (CCI) soil moisture (ESA CCI soil moisture, v08.1) provides satellite-retrieved surface soil moisture over the globe from a large set of active and passive microwave sensors (with soil penetration depths of ~2–5 cm). The dataset contains the following sub-products: "ACTIVE", "PASSIVE" and "COMBINED" (denoted ESA-CCI-ACT, ESA-CCI-PAS and ESA-CCI-COM in the following). The ESA-CCI-ACT and ESA-CCI-PAS products were created by using scatterometer (active microwave sensing) and radiometer (passive microwave sensing) soil moisture products, respectively. For ESA-CCI-COM, all active and passive single-sensor products are directly merged based on the signal-to-noise ratio of the input datasets (Gruber et al., 2019). ESA-CCI-COM outperforms the individual ESA-CCI-ACT and ESA-CCI-PAS products when compared to in situ soil moisture measurements (Gruber et al., 2019; Beck et al., 2021; Hirschi et al., 2023). Note that in the merging process for ESA-CCI-COM, the active and passive L2 products are scaled against surface soil moisture from the GLDAS-Noah v2.1 land surface model (Rodell et al., 2004) from which the dynamic range is inherited (Dorigo et al., 2017; Gruber et al., 2019). As of v08.1, a break-adjustment is implemented for ESA-CCI-COM, which corrects for breaks in mean and variance (Preimesberger et al., 2021; Su et al., 2016).

Microwave retrievals are impossible under snow and ice or when the soil is frozen, and complex topography, surface water, and urban structures have negative impacts on the retrieval quality (Dorigo et al., 2017; Dorigo et al., 2015). In addition, dense vegetation attenuates the microwave emission and backscatter from the soil surface and may (partly) mask the soil moisture signal. Altogether, these limitations result in spatial and temporal data gaps of remote sensing-based soil moisture estimates, with main affected areas in the high latitudes during winter and the tropical rainforests with very dense vegetation.

The product is provided on a 0.25° x 0.25° spatial grid and in daily temporal resolution from November 1978 onwards (in case of ESA-CCI-PAS and ESA-CCI-COM) or from August 1991 onwards, respectively (in case of ESA-CCI-ACT), and



data is available until 2022. Data coverage is limited in the early years of ESA-CCI-COM (and -PAS) when only few passive sensors are available (e.g., Loew et al., 2013). The inclusion of active sensors from July 1991 (Gruber et al., 2019) increased the spatio-temporal coverage. For this reason, certain applications of the product are often restricted to the period from 1992 130 onward (e.g., Nicolai-Shaw et al., 2017).

The ESA CCI soil moisture product has been extensively validated (Dorigo et al., 2015; Beck et al., 2021; Hirschi et al., 2023) and used in various research applications including monitoring climate variability and change, land atmosphere interactions, biogeochemical cycles and ecology, hydrological and land surface modelling, drought applications, as well as 135 (hydro)meteorological applications (see Dorigo et al., 2017 for an overview).

In addition to these ESA CCI surface soil moisture products, an ESA-CCI-COM-based root-zone soil moisture dataset is included in the analysis, which is derived by extrapolating surface soil moisture to deeper soil layers (denoted ESA-CCI-COM-RZSM). The extrapolation is based on an exponential filter (Wagner et al., 1999; Albergel et al., 2008), 140 which is applied to ESA-CCI-COM soil moisture (v08.0) and uses optimal values for the temporal length of the filter (T– parameter) determined from a large number of in situ time series (Pasik et al., 2023). The data represents the root zone down to one meter soil depth and will be released with the ESA CCI soil moisture products as of v09.1.

**2.1.2 C3S soil moisture**

The remote sensing dataset "Soil moisture gridded data from 1978 to present" (v202012.0.0 and v202012.0.1; Dorigo et al., 145 2021a) from the Copernicus Climate Data Store (CDS) provides estimates of surface soil moisture over the globe. This operational dataset uses a processing algorithm based on ESA CCI soil moisture version v05.2 for the product generation and the subsequent merging into the combined product (C3S, 2021), and considers near real-time L1 and L2 input datasets. Similar as for ESA CCI soil moisture, data based on ACTIVE and PASSIVE microwave remote sensing, as well as a COMBINED product are available (denoted C3S-SM-ACT, C3S-SM-PAS and C3S-SM-COM in the following). Compared 150 to ESA CCI soil moisture, C3S soil moisture is based on fewer sensors (and associated lower spatio-temporal coverage) and uses a consolidated processing algorithm that does not include the most recent developments of ESA CCI soil moisture v08.1 (e.g., no frozen ground cross-flagging, no seasonal rescaling, or no seasonal uncertainty estimation; Dorigo et al., 2023a). Also, for C3S-SM-ACT, the near real-time L2 dataset distributed by EUMETSAT is used instead of the H SAF product used by ESA CCI soil moisture (C3S, 2021).

155

The product is updated every ten days with a maximum latency of ten days. It is provided in 0.25° x 0.25° spatial and daily temporal resolution from November 1978 onwards (in case of C3S-SM-PAS and C3S-SM-COM) or from August 1991 onwards, respectively (in case of C3S-SM-ACT). The same limitations on coverage as for ESA CCI soil moisture apply in high latitudes during winter and the densely vegetated tropical regions.



### 2.1.3 ERA5

ERA5 is the fifth generation ECMWF reanalysis of the global climate and weather for the past decades (Hersbach et al., 2020). Data is available from 1940 onwards until present and updated daily with a latency of about 5 days. ERA5 is produced using 4D-Var data assimilation in CY41R2 of ECMWF's Integrated Forecast System (IFS). ERA5 provides hourly data with a spatial resolution of 31 km.

The land-surface scheme of ERA5, HTESSEL (Hydrology-Tiled ECMWF Scheme for Surface Exchanges over Land, Balsamo et al., 2009) distinguishes between four different soil layers with the following layer depths: layer 1 at 0–7 cm; layer 2 at 7–28 cm; layer 3 at 28–100 cm; and layer 4 at 100–289 cm. ERA5 is the first ECMWF reanalysis that includes remotely-sensed observations in a soil moisture analysis. Remote-sensing soil moisture from scatterometers (ERS-1,-2; MetOp-A,-B ASCAT) are assimilated in the land data assimilation from 1991 onward using a Simplified Extended Kalman Filter for the three soil moisture layers of the top first meter of the soil (Hersbach et al., 2020; De Rosnay et al., 2014). ERA5 soil moisture has been jointly evaluated with other reanalyses against in situ observations from various networks (Li et al., 2020; Beck et al., 2021). Compared with its predecessor ERA-Interim, ERA5 shows significant improvements in soil moisture.

On the one hand, we focus on layer 1 (0–7 cm), i.e., (near-)surface soil moisture to allow comparison with the C3S and ESA CCI remote sensing soil moisture products (see above). On the other hand, average soil moisture from layers 1–3 (i.e., 0–100 cm; layer-depth weighted) is considered as a representation of root-zone soil moisture. The ERA5 data has been re-gridded to a regular latitude/longitude grid of 0.25° x 0.25° for the CDS, where it is available in hourly temporal resolution. We have further aggregated the retrieved hourly data to daily means.

### 2.1.4 ERA5-Land

The land component of the ERA5 reanalysis provides global, hourly, high-resolution information of the water and energy cycles over land in a consistent representation (Muñoz-Sabater et al., 2021). ERA5-Land is a single simulation based on the land-surface model HTESSEL (Balsamo et al., 2009) forced by ERA5 near-surface atmospheric fields, with additional lapse-rate correction of temperature. Compared to ERA5, near-surface quantities are available in higher spatial resolution, and the soil parameters are more homogeneous between ERA5 production streams (Hersbach et al., 2020). There is no feedback from the land surface model to the atmospheric parameters, and atmospheric observations only influence the land surface simulations indirectly through the ERA5 forcing. Unlike ERA5, ERA5-Land does not assimilate remote sensing soil moisture or other land variables. ERA5-Land is available from 1950 onwards and updated monthly with a latency of about three months. It provides hourly data with a spatial resolution of 9 km, thus allowing more spatial detail compared to ERA5.



The representation of the soil compartments in ERA5-Land is consistent with ERA5 since both products consider the same land-surface model HTESSEL (see Sect. 2.1.3). Consequently, as for ERA5, soil moisture from layer 1 (surface soil moisture) and from layers 1–3 (root-zone soil moisture, layer-depth weighted average) are considered in the analyses.

Evaluation against in situ observations and other reference datasets shows the added value of ERA5-Land in the description of the hydrological cycle when compared to ERA5, with enhanced soil moisture and lake representation, and better agreement of river discharge with observations (Muñoz-Sabater et al., 2021). Soil moisture in particular shows a consistent improvement based on a large set of in situ observations (Beck et al., 2021; Muñoz-Sabater et al., 2021). The improvement is more marked for root zone soil moisture than for surface soil moisture.


The ERA5-Land data has been re-gridded to a regular latitude/longitude grid of 0.1° x 0.1° spatial resolution for the CDS, where it is available in hourly temporal resolution. The aggregation of the hourly data to daily means has been done using the ERA5 daily statistics calculator of the CDS[1].

**2.1.5 MERRA-2**

The Modern-Era Retrospective Analysis for Research and Applications, version 2 (MERRA-2), is the latest atmospheric reanalysis of the modern satellite era produced by NASA's Global Modeling and Assimilation Office (GMAO; Gelaro et al., 2017). It was introduced to replace the original MERRA dataset because of the advances made in the assimilation system that enables assimilation of modern hyperspectral radiance and microwave observations, along with GPS-Radio Occultation datasets. Among the advances in MERRA-2 are the assimilation of aerosol observations, several improvements to the

representation of the stratosphere including ozone, and improved representations of cryospheric processes. Other improvements in the quality of MERRA-2 compared with MERRA include the reduction of some spurious trends and breaks related to changes in the observing system and reduced biases and imbalances in aspects of the water cycle. MERRA-2 provides data beginning in 1980, at 0.625° x 0.5° spatial resolution and hourly temporal resolution. For an overview on the dataset, see Gelaro et al. (2017).


The land surface model used in MERRA-2 is the Catchment model (CLSM; Koster et al., 2000). It explicitly addresses subgrid-scale soil moisture variability and its effect on runoff and evaporation, using the basic computational element of a hydrological catchment. The land hydrology of MERRA-2 has been assessed against GRACE terrestrial water storage data as well as against in situ soil moisture data (Reichle et al., 2017b). MERRA-2 is produced using four separate streams,

initialised in 1979, 1991, 2000, and 2010. The first year of each stream is designated as spinup (Bosilovich et al., 2015). The land surface restart files for each MERRA-2 stream were themselves spun up for at least 20 years, using the offline (land only) version of the MERRA-2 land model forced with MERRA surface meteorological fields (Reichle et al., 2017a).

---

[1] https://cds.climate.copernicus.eu/cdsapp#!/software/app-c3s-daily-era5-statistics



Despite this allowance for a spinup, it has been documented that discontinuities remain in the high latitudes for root-zone soil moisture (cf. Fig. 13 of Reichle et al., 2017a).


The variables SFMC (water surface layer, 0–5 cm depth) and RZMC (water root zone, 0–100 cm depth) have been retrieved from the Goddard Earth Sciences Data and Information Services Center (GES DISC) as daily aggregated data (GMAO, 2015).

**Table 1 Product summary of the considered soil moisture products. The column 'horizontal grid spacing' lists the resolution of the data retrieved, if applicable the native resolution of the dataset is listed in brackets. Similarly, for 'temporal resolution', values between brackets indicate the underlying temporal resolution when this differs from the retrieved resolution.**

| Dataset | Institution | Type of product | Time range | Horizontal grid spacing | Soil layer depth | Variable name | Temporal resolution | Main reference |
|---|---|---|---|---|---|---|---|---|
| ESA-CCI-COM/-ACT/-PAS v8.1 | ESA | Active and passive microwave remote sensing | Nov 1978 / Aug 1991–2022 | 0.25°x0.25° | ~2–5 cm | sm | Daily | Gruber et al. (2019); Dorigo et al. (2017); Dorigo et al. (2023b) |
| ESA-CCI-COM-RZSM | TU Wien | Exponential filter-based root-zone soil moisture | 1991–2021 | 0.25°x0.25° | 0–100 cm | rzsm_1m | Daily | Pasik et al. (2023) |
| C3S-SM-COM/-ACT/-PAS v202012 | Copernicus | Active and passive microwave remote sensing | Nov 1978 / Aug 1991–present | 0.25°x0.25° | ~2–5 cm | sm | Daily | C3S (2021); Dorigo et al. (2021a) |
| ERA5 | ECMWF | Atmospheric reanalysis | 1940–present | 0.25°x0.25° (~31 km) | 0–7 cm, 0–100 cm | swvl1, swvl1–3 | Hourly | Hersbach et al. (2020) |
| ERA5-Land | ECMWF | Land-surface reanalysis | 1950–present | 0.1°x0.1° (~9 km) | 0–7 cm, 0–100 cm | swvl1, swvl1–3 | Daily (hourly) | Muñoz-Sabater et al. (2021) |
| MERRA-2 | NASA | Atmospheric reanalysis | 1980–present | 0.625°x0.5° | 0–5 cm, 0–100 cm | SFMC, RZMC | Daily (hourly) | Bosilovich et al. (2015) |

## 2.2 Other variables from reanalyses and observations

Apart from soil moisture, the following variables are used from the ERA5, ERA5-Land and MERRA-2 reanalysis products: total precipitation, evapotranspiration, runoff and surface air temperature. Note that ERA5 and ERA5-Land share the same precipitation data, except for the higher spatial resolution of the latter.

Observed daily global land-surface precipitation totals based on data of national meteorological and hydrological services,
regional and global data collections as well as WMO GTS-data are provided within the GPCC Full Data Daily Product





Version 2022 (Ziese et al., 2022). The data is available at a regular latitude/longitude grid with a spatial resolution of 1° x 1° degree and covers the 1982–2020 period.

Daily global gridded temperature anomaly fields (w.r.t. 1951–1980 climate) based on various sources of station observations

are taken from Berkeley Earth (Rohde et al., 2013). Data is available at a spatial resolution of 1° x 1° latitude/longitude grid, covering the period 1880-present.

### 2.3 Land-surface characteristics and bioclimatic indicators

ESA CCI provides a set of ancillary datasets that were used in the generation of the products (Dorigo et al., 2021b). The ESA CCI soil porosity map has been used to convert ESA-CCI-ACT and C3S-SM-ACT from the original units "percentage

of saturation" to volumetric soil moisture in $m^3$ $m^{-3}$ as provided by the other considered products. The porosity map has been derived according to Saxton and Rawls (2006) taking clay, sand, silt, and organic matter of the Harmonized World Soil Database as input.

For investigating controls of the soil moisture trends based on the various products (see Sect. 3.3), these are compared

globally to those calculated using the maximum data availability over the 2000–2020 period for Vegetation Optical Depth (VOD; Moesinger et al., 2020), ERA5-derived aridity (2000–2018 only; Wouters, 2021), fractional inundated area (Schroeder et al., 2015), and fractional covers of urban area, of bare soil and of tree cover (C3S, 2019).

## 3 Methods

### 3.1 Event definitions

Based on guidance from the WMO (2016), extreme weather and climate events can be described quantitatively by a combination of the following metrics and information:

- An Index describing the anomaly from normal conditions (based on observations)
- A Threshold (above or below which conditions become 'extreme')
- Temporal information (records of the start date, end date, and duration)
- Spatial information (geographic area affected)

Here we focus on documented major drought events of the past two decades, with regions and periods that are predefined based on scientific literature and drought reports. From 2011 onward, in particular the Bulletin of the American Meteorological Society (BAMS) "Explaining Extreme Events from a Climate Perspective" report series is considered for

this purpose. These event definitions serve as spatial and temporal bounds for the characterisation of the individual drought events. An overview on the considered events and their predefined event regions and event periods is given in



Supplementary Table 1. This analysis extends on the extreme event catalogue and event metrics developed within the C3S_511 (Crezee et al., 2019; Yang et al., 2022).

## 3.2 Index and drought metrics calculation

The soil moisture products considered here are given in volumetric moisture content (i.e., in units of m$^3$ m$^{-3}$), except for ESA-CCI-ACT and C3S-SM-ACT which are originally in percent of saturation and converted to volumetric moisture content using the ESA CCI soil porosity map (see above). However, care needs to be taken when comparing absolute soil moisture from different sources since the absolute values are known to be dependent on underlying assumptions of the land-surface models and related soil property datasets (e.g., differing soil depths and soil properties like porosity) as well as on

varying penetration depths of the remote-sensing products. We apply a standardization (Z–transformation) to remove differences in absolute levels and variability of the soil moisture values between the products, but also between locations, and focus on the temporal anomalies (see e.g., Koster et al., 2009; Orlowsky and Seneviratne, 2013).

These unitless standardised soil moisture anomalies (e.g., Orlowsky and Seneviratne, 2013) are based on daily input data and

are calculated per individual grid point with respect to the climatology of the 2000–2020 reference period:

$$SMA_{y,d} = \frac{SM_{y,d} - \mu_{\mathrm{d}}}{\sigma_{\mathrm{d}}} \qquad\qquad\qquad (1)$$

In Eq.1, $SM_{y,d}$ denotes soil moisture at any year $y$ and day $d$, while $\mu_{\mathrm{d}}$ and $\sigma_{\mathrm{d}}$ denote the climatological mean and inter-

annual standard deviation of soil moisture of day $d$ calculated over the reference period. To enhance the sample size, calculation of $\mu_{\mathrm{d}}$ and $\sigma_{\mathrm{d}}$ for each day of the year is based on the days in a 11-day window around $d$, i.e., by applying a 11-day moving average on the original soil moisture time series. Note that a reference period identical to the analysis period is chosen in order to keep the calculation of the standardised anomalies independent of temporal fluctuations in the original time series prior to that, and also to avoid reduced data coverage in the remote sensing products in earlier periods (e.g.,

Hirschi et al., 2023). A 3-day running mean is applied in addition on the resulting daily standardised anomalies with the purpose to fill daily gaps in the remote-sensing products.

For the definition of a drought, a threshold value of –1.5 standardised anomaly is chosen and any value below this is considered being in an abnormal dry state (i.e., $SMA_{y,d} < -1.5$). This threshold is inspired by the SPI-based categorization

of droughts, where values below –1.5 represent severe to extreme drought (Mckee et al., 1993).

Four different metrics are defined for characterizing each drought event within the predefined event region and event period (see Supplementary Table 1). The *magnitude* of the event is the minimum standardised anomaly over time. The *duration* of





the event is the total number of days (not necessarily consecutive) during the event for which the standardised anomaly is

below the threshold value of –1.5. The *severity* is defined as the time accumulated standardised anomalies over the days (i.e., consecutive, and non-consecutive) for which the standardised anomaly is below the threshold value and is given in units of days times unitless standard deviations (i.e., days*1). These metrics are all calculated on the grid point scale. In addition, the temporally varying *spatial extent* of the event is defined as the area in which the standardised anomaly is below the threshold of –1.5.


Note that the application of the 3-day running mean smoothing of the standardised anomalies helps to fill temporal gaps in the remote sensing products, while still being as close as possible to the original data. For the remote sensing products, a larger smoothing window size (i.e., 5-day window) mainly impacts the calculation of the magnitude, while duration and severity only slightly change. Also, sensitivity tests showed that there is not much impact of varying smoothing windows on

the results for the reanalysis products (not shown).

Since the severity captures both the duration and the amplitude of the event, it is suitable for defining the most affected *core* of the event region as represented by the products. This core region is defined as all grid points for which the severity is larger than the median of all non-zero severity grid points of the event and is used to spatially aggregate the drought metrics

for summarising the events.

**3.3 Dry-season trend estimation**

The analysis of dry-season trends is based on the 25% climatologically driest days of the year based on the ERA5-Land surface soil moisture climatology. ERA5-Land is chosen here since it has shown better skill than the other considered products when compared to in situ observations (Li et al., 2020; Beck et al., 2021). As defined in the previous section, the

mean climatology is calculated based on smoothed time series by applying a 11-day moving average on the original soil moisture data. The soil moisture climatology is further masked for negative soil temperatures (using ERA5-Land soil temperature from layer 1) before extracting the dry season. The dry-season data are yearly averaged, and trends are derived using the Theil-Sen trend estimator. Significance of the trends is determined with the Mann-Kendall test with a false rejection rate (or alpha value) of 0.05, and non-significant trends are masked for display of the trend maps when indicated.


Apart from surface and root-zone soil moisture, also dry-season trends in other relevant variables of the land water balance (i.e., total precipitation, evapotranspiration, runoff, and surface air temperature), as well as in daily data of VOD and inundated area, are considered by applying the described dry-season mask based on surface soil moisture of ERA5-Land. For an easier comparison to these variables, we focus on trends in absolute soil moisture.



## 4 Results

Detailed results are presented for the recent 2022 drought in Western-Central Europe (Schumacher et al., 2022; Schumacher et al., 2023) as an example. This event serves as a showcase for the application of the near real-time products (i.e., products with a latency of less than one year are considered; these include the satellites products C3S-SM-COM, C3S-SM-ACT, C3S-SM-PAS, and the reanalysis products ERA5, ERA5-Land and MERRA-2). This is followed by the characterization of multiple recent major drought events worldwide considering all products.

### 4.1 Drought Europe 2022

The severity of the 2022 drought event appears largest in central and western Europe, with highest values based on surface soil moisture in Germany, Switzerland, France, Italy as well as parts of eastern Europe (Fig. 1 a–f). This core of the event region (see Methods) is captured by all products, but the region appears more coherent in the reanalysis products as compared to the remote sensing products (C3S-SM-COM, -ACT, and -PAS). Due to the strong correlation of surface soil moisture with deeper soil layers (e.g., Hirschi et al., 2014), the location of the event in the root-zone is very similar compared to the surface layer (Fig. 1 g–i). However, the core region is slightly less coherent and widespread in the root zone (cf. also horizontal line segments of Fig. 5, which indicate the spatial extent of the core region).

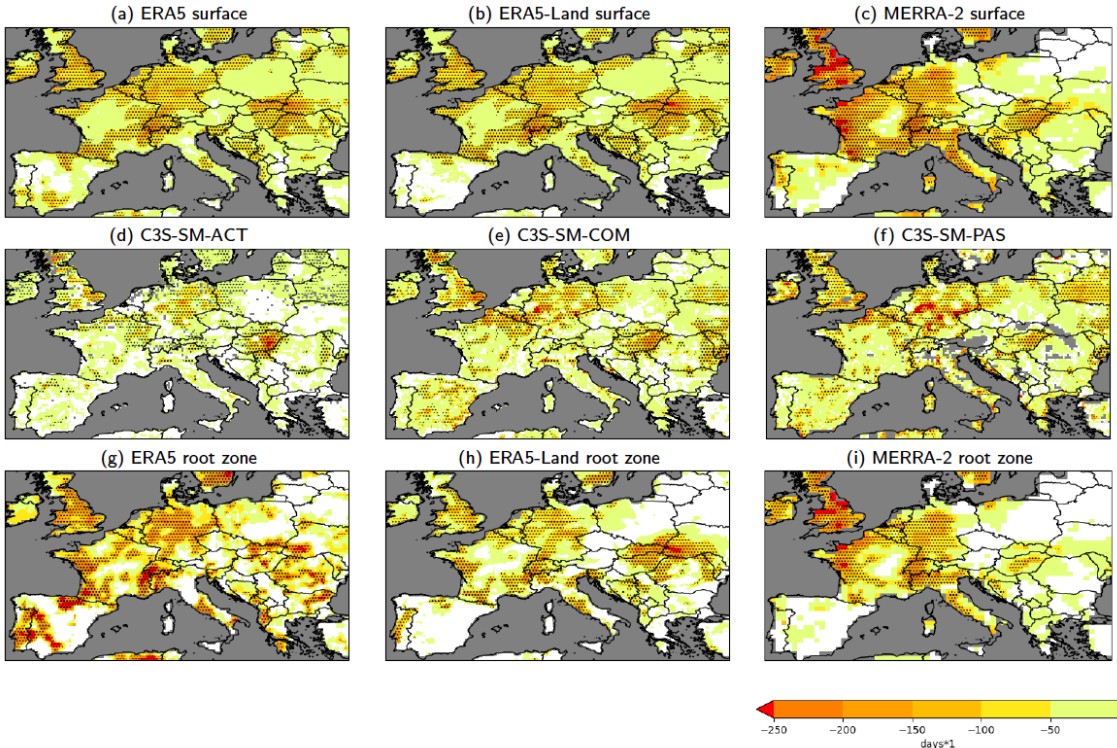

**Figure 1 Severity of the 2022 Europe drought event based on the time accumulated soil moisture anomalies in (a–f) the surface and (g–i) the root zone layer. The core of the event region is stippled.**



For surface layer soil moisture, the event appears most severe in MERRA-2 (though within a smaller extent of the core region than the other reanalyses), followed by C3S-SM-PAS, ERA5-Land and ERA5 (see also Table 2). C3S-SM-COM and

in particular C3S-SM-ACT show weaker severities for this event. The magnitude of the 2022 European drought based on C3S-SM-PAS/-COM is over large parts comparable to the reanalysis products, with standardised anomalies of –3 and less in parts of the core region of the event (Fig. 2, Table 2). C3S-SM-ACT shows weaker magnitudes (see also Table 2). The event shows the longest durations in MERRA-2, with over 90 days in parts of the core region and over 50 days on the average over it (Fig. 3 and Table 2). ERA5/ERA5-Land and C3S-SM-PAS/-COM display average durations of 30 to 40 days, and

C3S-SM-ACT the shortest with about 20 days.

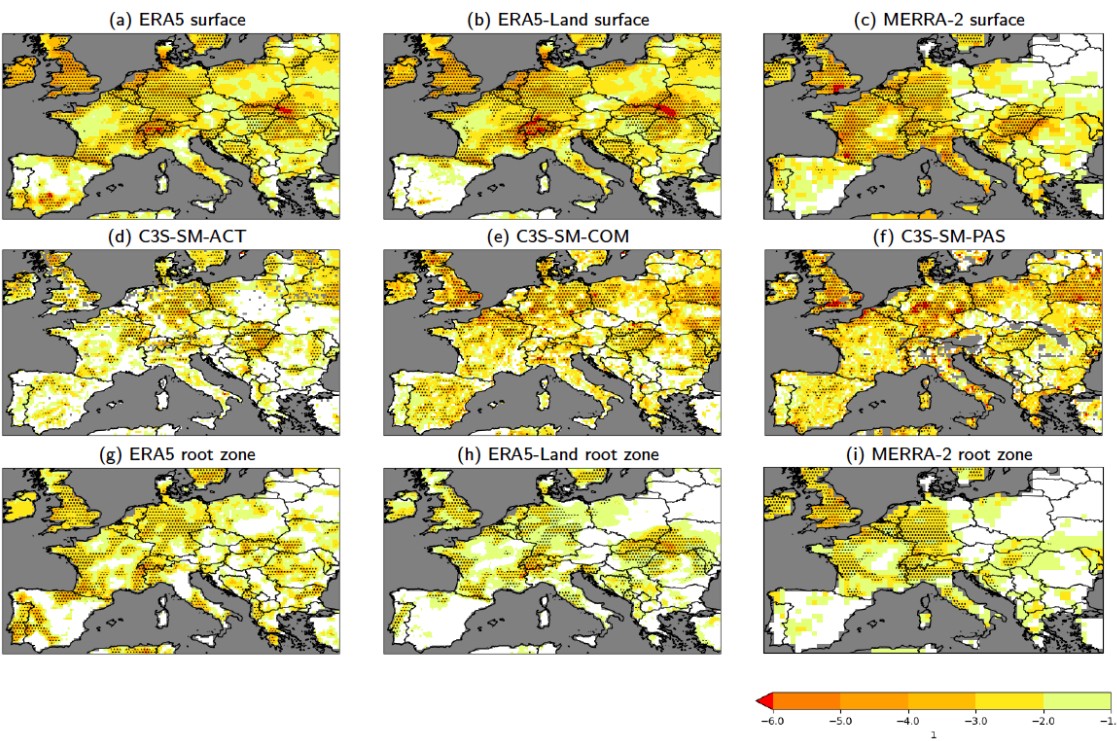

**Figure 2 Magnitude of the 2022 Europe drought event based on the temporal minimum of the standardised soil moisture anomalies in (a–f) the surface and (g–i) the root zone layer. The core of the event region is stippled.**



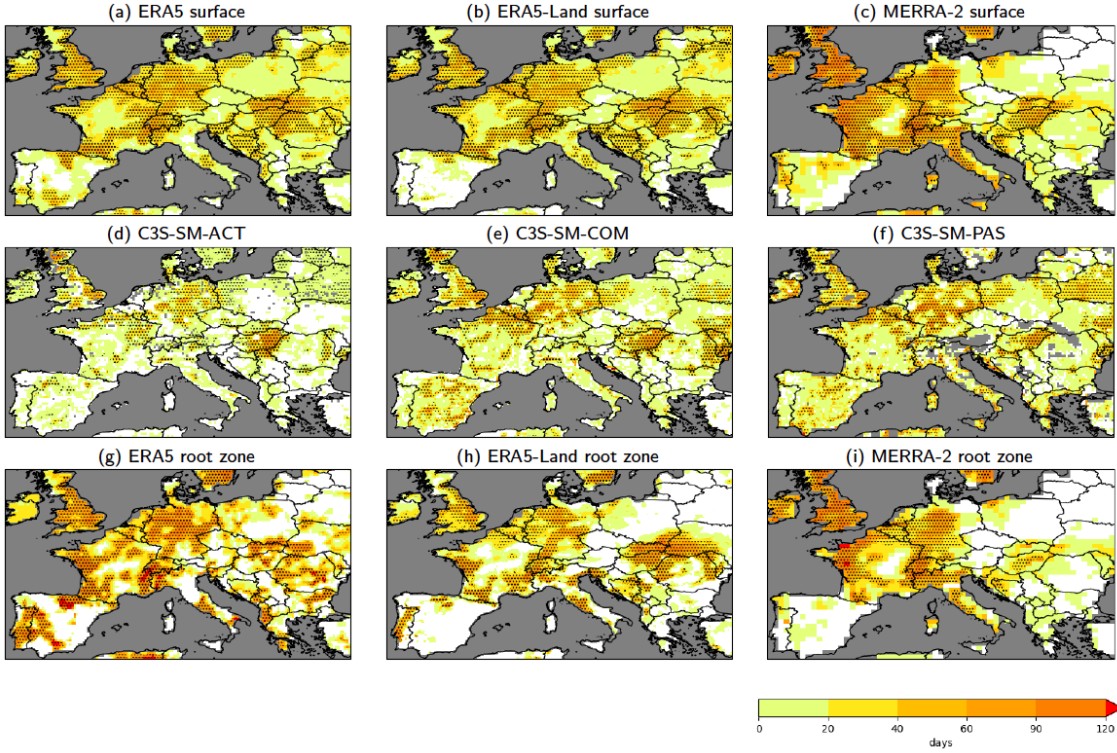


**Figure 3 Duration of the 2022 Europe drought event based on the number of days within the event period with standardised soil moisture anomalies below –1.5 in (a–f) the surface and (g–i) the root zone layer. The core of the event region is stippled.**

In the root-zone of ERA5-Land and particularly of ERA5, the 2022 European drought appears more severe as compared to the surface layer (Fig. 1 g–i, Table 2), mostly due to longer durations (Fig. 3 g–i), while the magnitudes are weaker

(Fig. 2 g–i). MERRA-2 shows a similarly reduced drought magnitude in the root zone as ERA5/ERA5-Land, but in contrast, the duration is only marginally increased compared to the surface layer (where it is however already substantially longer than in ERA5/ERA5-Land, Table 2). Together with weaker negative anomalies (cf. Fig. 4), the severity in the root zone of MERRA-2 becomes slightly reduced compared to the surface layer.

The temporal evolution of the standardised surface soil moisture anomalies averaged over the core of the event region shows two strongest phases of the event in the second half of July and first half of August 2022 (Fig. 4), with average standardised anomalies based on the reanalysis products below –1.5. Anomalies are most pronounced (i.e., below –2) and prolonged for MERRA-2 compared to the other products. ERA5 and ERA5-Land show slightly weaker negative anomalies and a quicker return to normal conditions. Among the remote-sensing products, C3S-SM-PAS shows the most negative anomalies,

however only with the minimum in the first half of August. Over the whole period, C3S-SM-COM and particularly C3S-SM-ACT show overall weakest standardised anomalies, which are on the average only temporary below normal conditions.





Root-zone soil moisture anomalies from the reanalysis products display less temporal variation, with longer-lasting
minimum values in July and August (dashed lines in Fig. 4). In the root zone, the dryer than normal conditions persist during
late summer and early autumn, with values still below –1 standardised anomaly. Strongest and most prolonged root-zone soil
moisture anomalies are displayed by ERA5, while ERA5-Land and MERRA-2 are comparable.

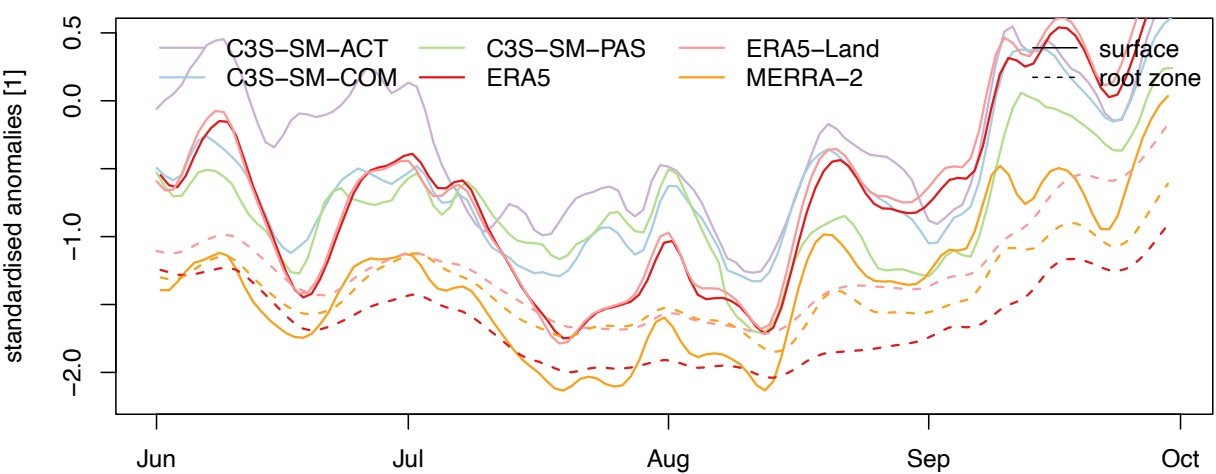

**Figure 4 Average of the standardised surface (full lines) and root-zone (dashed lines) soil moisture anomalies within the respective
core of the event regions (cf. stippled regions in Figs. 1–3) during the 2022 Europe drought event period.**

The temporal evolution of the spatial extent of the drought (i.e., the area of standardised soil moisture anomalies below –1.5
within the core of the event region) shows highest values during July, reaching 1.3 Mio. $km^2$ for surface soil moisture of
ERA5 and MERRA-2 (Fig. 5, Table 2), followed by 1.23 Mio. $km^2$ of ERA5-Land. The spatial extent of the event is
considerably smaller in the remote sensing products and reaches around 0.9 Mio $km^2$ for C3S-SM-PAS, and even less for the
other two products. The spatial extent of the event in the root-zone is lower compared to the respective surface layer in
ERA5-Land and MERRA-2, while it is only slightly reduced in ERA5. As for the averaged standardised anomalies, ERA5
appears separated from ERA5-Land and MERRA-2, with a maximum spatial extent of 1.26 Mio. $km^2$ as compared to about
0.8 to 0.9 Mio. $km^2$ of the other products (Table 2).


Based on the involved products, the 2022 drought event is characterised by a severity ranging from –124.9 to –45.8 days*1
in the surface layer, respectively –154.9 to –97.4 days*1 in the root zone, a magnitude ranging from –3.4 to –2.6,
respectively –2.6 to –2.2, and a duration ranging from 22 to 57 days, respectively 52 to 75 days (Table 2).



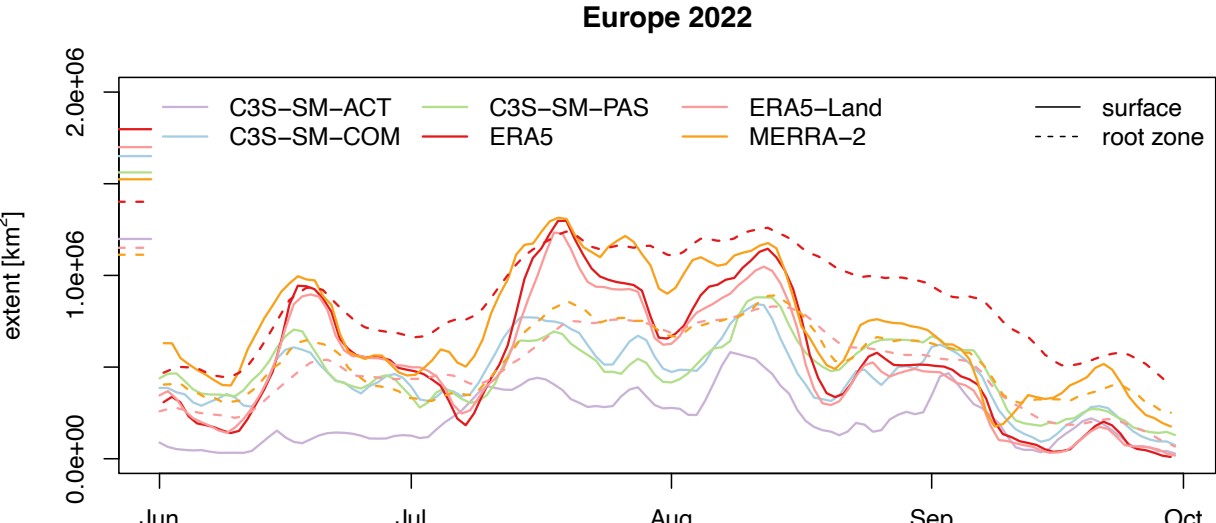

**Figure 5** Spatial extent of the standardised surface (full lines) and root-zone (dashed lines) soil moisture anomalies below –1.5 within the core of the event region during the 2022 Europe drought event (full lines). Line segments at the left border of the figure indicate the extent of the core region for each dataset (i.e., corresponding to the extents of the stippled areas of Figs. 1–3 where the severity is larger than the median of all non-zero severity grid points).

**Table 2** Overview on characteristics of the 2022 Europe drought event as represented by surface and root-zone soil moisture of the different products. The metrics represent the area mean over the respective core of the event region for severity, magnitude and duration, and the temporal maximum for the spatial extent of the event. Also noted is the mean of these metrics based on all products.

|  | Dataset | Severity [days*1] | Magnitude [1] | Duration [days] | Extent [1e6 km²] |
|---|---|---|---|---|---|
| **Surface soil moisture** | C3S-SM-ACT | –45.8 | –2.6 | 22 | 0.58 |
|  | C3S-SM-COM | –72.0 | –3.1 | 33 | 0.84 |
|  | C3S-SM-PAS | –84.1 | –3.4 | 37 | 0.88 |
|  | ERA5 | –78.8 | –3.0 | 37 | 1.30 |
|  | ERA5-Land | –80.8 | –3.1 | 36 | 1.23 |
|  | MERRA-2 | –124.9 | –3.1 | 57 | 1.31 |
|  | *Product range* | *[–124.9,–45.8]* | *[–3.4,–2.6]* | *[22,57]* | *[0.6,1.3]* |
| **Root-zone soil moisture** | ERA5 | –154.9 | –2.6 | 75 | 1.26 |
|  | ERA5-Land | –97.4 | –2.2 | 52 | 0.83 |
|  | MERRA-2 | –113.2 | –2.3 | 59 | 0.89 |
|  | *Product range* | *[–154.9,–97.4]* | *[–2.6,–2.2]* | *[52,75]* | *[0.8,1.3]* |



## 4.2 Recent major drought events

An overview on the characteristics of major drought events in the 2000–2020 period globally as represented in the various products is given in Figs. 6–9 (based on surface and root-zone soil moisture). For the remote-sensing products, we focus on the COMBINED products for better readability (corresponding results for the ACTIVE and PASSIVE products can be found in the Supplementary Figs. 1–4).

Among the seasonal, sub-annual events, the Iberian Peninsula 2011–2012 drought and the Texas 2011 and Great Plains 2012 droughts appear most severe and longest in most of the products. In terms of the drought magnitude, the differences between the events are less pronounced. In the surface layer, the east African drought of 2015 shows strongest magnitudes (except in MERRA-2), followed by Iberian Peninsula 2011–2012 drought. The spatial extents of the drought reach highest values for the Europe 2020 and the South Africa 2015–2016 droughts, though with large differences between the products. The multi-year droughts are not directly comparable since they consist of multiple phases of dryness with intermittent recoveries.

As for the 2022 drought event in Europe, ESA-CCI-/C3S-SM-COM partly show weaker drought severities compared to the reanalysis products (Fig. 6). It is evident for various events in Europe (e.g., Europe 2018, Europe 2020), and for the events in Middle East and the western North America. Among the remote sensing products, the ACTIVE products often display reduced drought severities compared to the COMBINED products (Supplementary Fig.1). This is to a lesser extent also the case for the PASSIVE products. The partly weaker severities of these events in ESA-CCI-/C3S-SM-COM are both related to shorter event durations (Fig. 7) as well as often less pronounced and less continuous average negative anomalies (see e.g., Fig. 4 for the 2022 drought in Europe). Also, the spatial extent of these events is partly reduced in the remote-sensing products in these cases (Fig. 9). For other events in Europe (e.g., Europe 2003, Europe 2013), and for the events in Western Russia and Africa, ESA-CCI-/C3S-SM-COM show similar or partly stronger severities than the reanalysis products (and corresponding similar or longer durations). For the magnitudes, in particular MERRA-2 often shows weaker signals compared to the other products (Fig. 8), while the durations are partly prolonged and the droughts more severe (Figs. 6–7). Also, the ACTIVE remote-sensing products show mostly weaker drought magnitudes compared to the other products (Supplementary Fig. 3).

In the root zone, the drought events appear weaker in terms of magnitudes and often smaller in the spatial extents, and in some cases prolonged. Compared to the reanalysis products, the ESA-CCI-COM-based root-zone soil moisture (ESA-CCI-COM-RZSM) shows partly stronger drought severities and corresponding longer event durations (e.g., Iberian Peninsula 2011–2012, Europe, 2018, Great Plains 2012), as well as often stronger magnitudes. Also, the spatial extents of the droughts appear larger in some cases in ESA-CCI-COM-RZSM (e.g., Southeast Europe 2007, South Africa 2015–2016, Great Plains 2012).





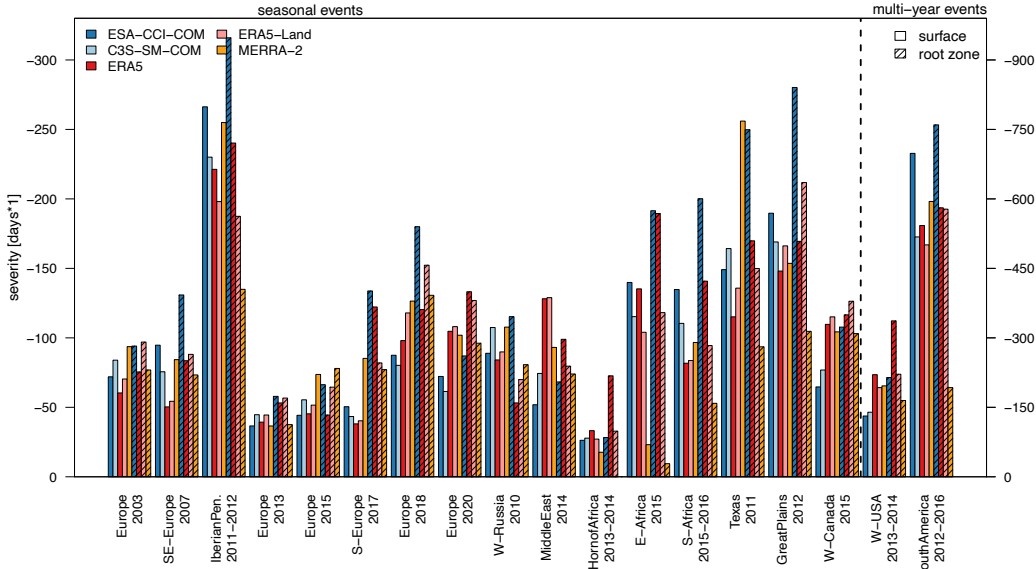

**Figure 6 Severity of recent major drought events. The values are based on surface soil moisture (empty bars) and root-zone soil moisture (dashed bars) and represent the area mean over the respective core of the event region. Note the righthand y-axis for multi-year events. Corresponding results for the ACTIVE and PASSIVE remote sensing products in relation to the COMBINED products can be found in the Supplementary Information.**

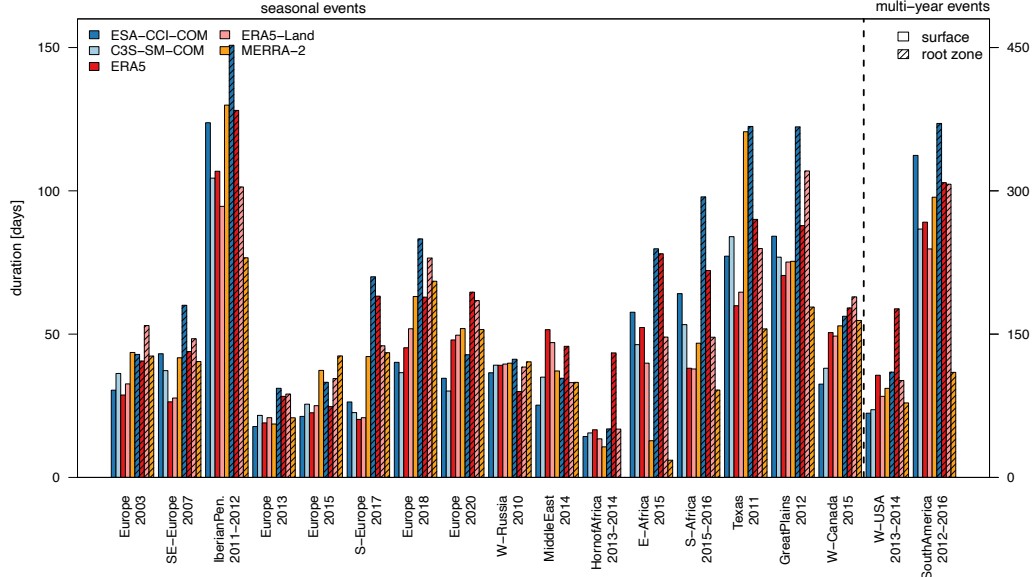

**Figure 7 As Fig. 6, but for the duration of recent major drought events.**






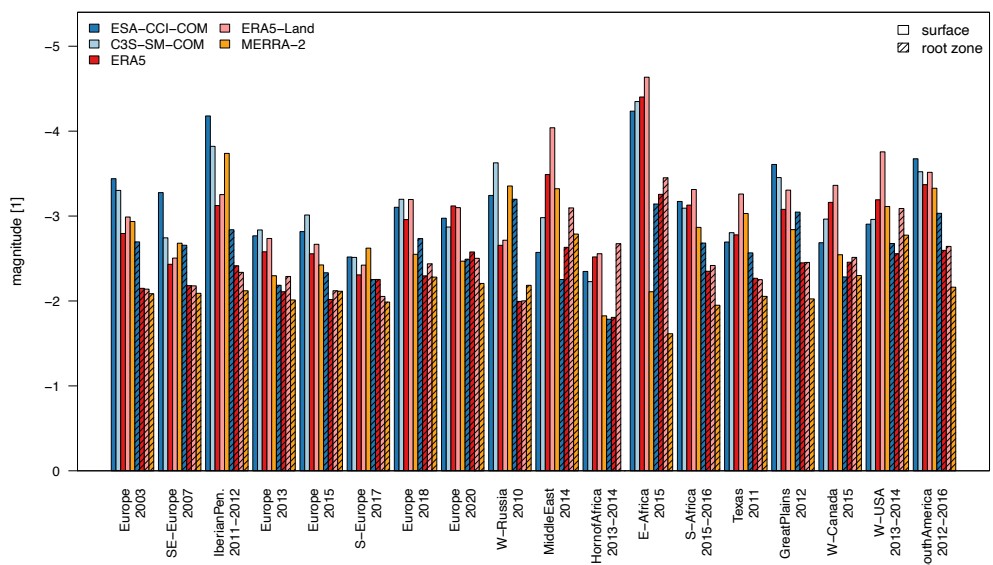

**Figure 8 As Fig. 6, but for the magnitude of recent major drought events.**

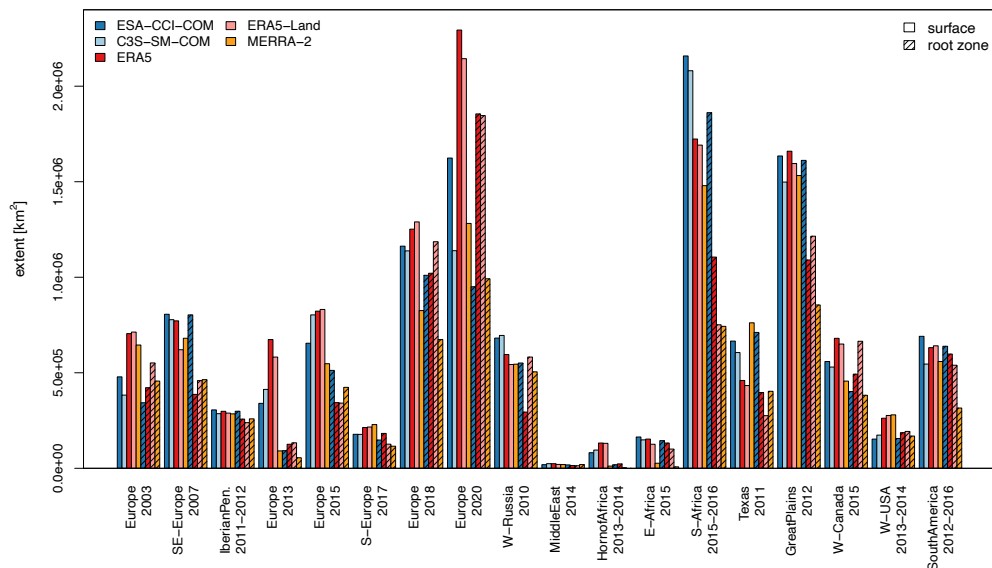

**Figure 9 As Fig. 6, but for the spatial extent of recent major drought events. In this case, the values represent the temporal maximum of the spatial extent (with logarithmic scale) of the events.**





## 4.3 Product intercomparison

The overall product behaviour during the analysed drought events is summarised in Fig. 10. In line with the results of the previous section, the dampened drought magnitudes and smaller spatial extents in the root zone compared to the surface

layer are again visible in the respective products. Also, a tendency for prolonged durations of the droughts in the root zone is observable (except for MERRA-2, which already shows partly longer durations in the surface layer compared to the other products). ESA-CCI-COM-RZSM displays partly stronger representation of the droughts in all metrics compared to the reanalysis products, while MERRA-2 shows weaker drought magnitudes and partly shorter durations and lower severities.

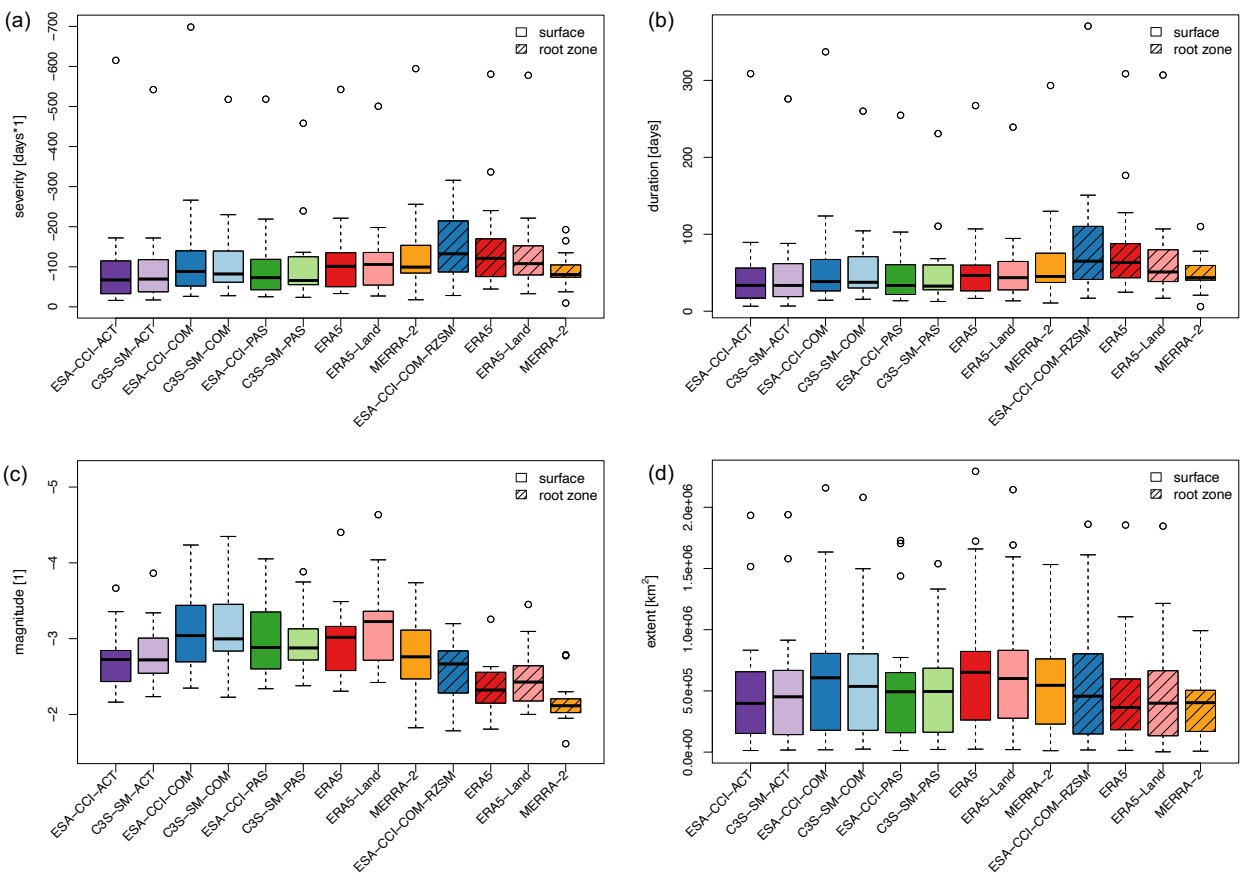


**Figure 10 Product intercomparison based on the drought metrics (a) severity, (b) duration, (c) magnitude, and (d) spatial extent. The box-whisker plots represent the distributions of the drought metrics for the analysed 18 drought events.**

For surface soil moisture, the ACTIVE remote sensing products, and to a lesser extent the PASSIVE products, tend to show

weaker drought signals in all presented metrics compared to the COMBINED products and ERA5/ERA5-Land. This is most pronounced for the drought magnitudes of ESA-CCI-/C3S-SM-ACT. As for the root zone, also MERRA-2 displays weaker drought magnitudes compared to the other products, while the droughts appear partly prolonged and more severe in its



surface layer (cf. also the 2022 Europe drought above and corresponding Figs. 3–4, and Table 2, as well as Fig. 7 for all events).


The spatial extents of the droughts based on surface soil moisture tend to be larger for the reanalysis and the COMBINED remote sensing products. In the other remote sensing products, the spatial extents of larger droughts appear smaller.

**4.4 Trends in dry-season soil moisture**

To understand differences in the representation of the droughts by the analysed products, their trends in dry-season soil
moisture over the 2000–2020 period are analysed in the following. ERA5 and ERA5-Land show mostly significantly negative dry-season trends, particularly for surface soil moisture (Fig. 11, cf. also Table A1 in Appendix A). Strong drying can be observed for the region north of the Black Sea/Caspian Sea, the Amazon region, central and southern Africa, and parts of Asia, Europe and Australia. MERRA-2 in contrast displays widespread significantly positive dry-season trends in both surface and root-zone soil moisture, except for the Amazon region, southern Africa, and Australia, where consistent
negative trends are observable in all reanalysis products. Consistent positive trends in the reanalyses can be observed in east Asia, central north America, and in north-western Russia (surface soil moisture only), but these appear less widespread in ERA5/ERA5-Land compared to MERRA-2. Trends in ESA-CCI-/C3S-SM-COM, as well as ESA-CCI-COM-RZSM appear significantly negative in the Black Sea/Caspian Sea region, southern Africa, parts of South America and Australia, and in Siberia, and thereby mostly agree with ERA5/ERA5-Land. Positive trends in the COMBINED remote-sensing products in
parts of east Asia and north America tend to agree more with MERRA-2.

Within the remote-sensing products, ESA-CCI-/C3S-SM-COM and ESA-CCI-PAS show more widespread negative dry-season trends (Fig. 14, cf. also Table A1). In contrast, the ACTIVE products and C3S-SM-PAS and display larger fractions and more pronounced positive trends. Partly consistent negative trends can be observed in parts of Siberia, the Black
Sea/Caspian Sea region, southern Africa and parts of central Europe, consistently positive trends mainly in southeast Asia, parts of north-western Russia and eastern USA.

Of all products, ESA-CCI-COM, ERA5-Land and ERA5 show largest area fractions of negative surface soil moisture trends (70.3, 61.6 and 61.5 % respectively, trends not masked for significance), while MERRA-2, C3S-SM-PAS and -ACT show
largest fractions of positive trends (63.6, 59.5 and 58.9 % respectively; see Table A1). Similarly, largest area fractions of negative root-zone soil moisture trends are present in ESA-CCI-COM-RZSM and ERA5 (66.3 and 62.2 %), and largest fractions of positive trends in MERRA-2 (64.4 %). Products that show larger area fractions of positive than negative trends (i.e., MERRA-2, C3S-SM-ACT/-PAS, and ESA-CCI-ACT) also tend to display less pronounced drought magnitudes (cf. Fig. 10). Nevertheless, some regions with largely consistent drying trends are apparent in parts of central Europe, in a region



north of the Black Sea/Caspian Sea, in southern Africa, and in parts of Australia, Siberia and South America. These regions are mostly consistent with previous studies on trends in dry-season water availability (e.g., Padron et al., 2020)

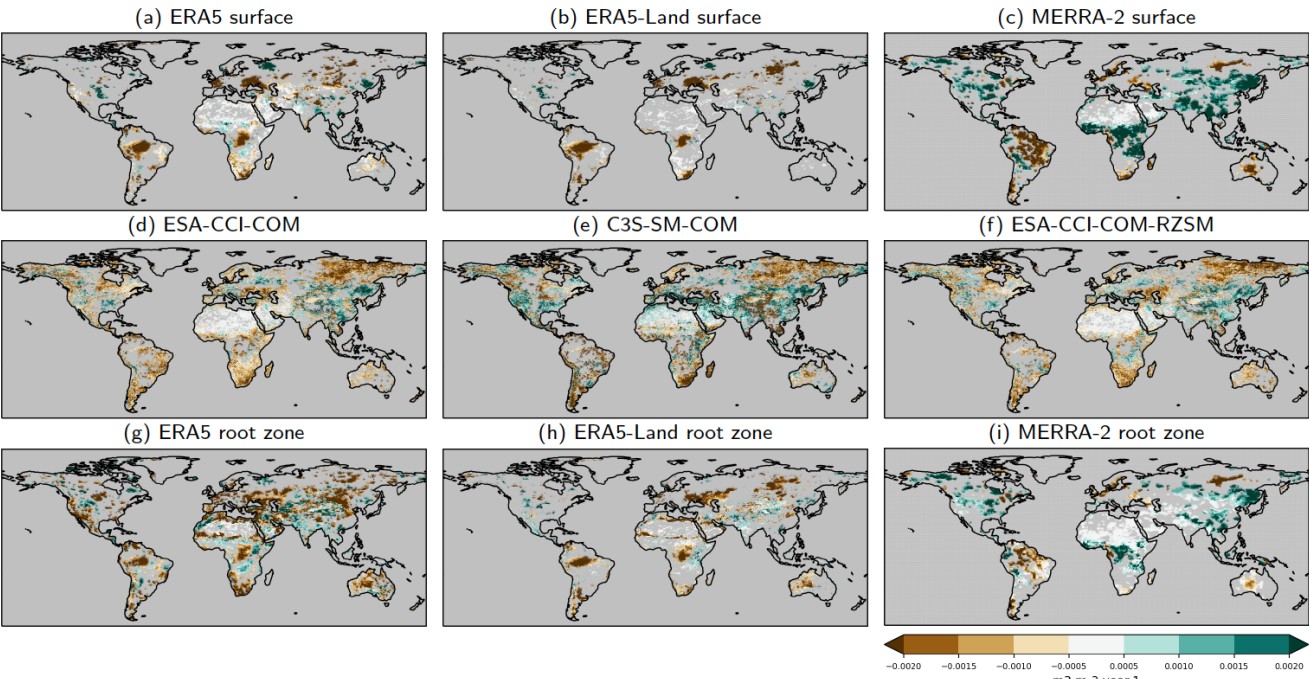

**Figure 11 Theil-sen trend estimate (m³ m⁻³ yr⁻¹) on yearly mean dry-season soil moisture (i.e., 25% climatologically driest days of**
**the year based on ERA5-Land surface soil moisture) in (a–e) the surface and (f–i) the root zone layer, 2000–2020 period. A Mann-Kendall test with a false rejection rate (or alpha value) of 0.05 was performed to mask out regions where no significant trend is present.**

## 5 Discussion

### 5.1 Relation of drought representation and dry-season soil moisture trends

In the following, we investigate the effect of the diverse and partly contradictory dry-season soil moisture trends of the products on their representation of the investigated drought events. For each event and product, Fig. 12 shows the product deviation in the representation of these events in terms of severity and magnitude versus the deviation in the 2000–2020 trend. Trends and drought responses are averaged over the respective core region of the events, and the respective deviations are calculated with respect to the product median of the individual events. The chronology of the events within the 2000–
2020 period is indicated with increasing circle sizes.

Despite the considerable spread of the products around the product medians of the trend and drought severity respectively magnitude, the scatter plots reveal that products with larger deviations in the trends (i.e., stronger negative or more positive



trends) are connected with stronger respectively weaker drought severity and magnitude (i.e., corresponding to negative respectively positive deviations in these two drought metrics). This relation appears more pronounced for drought severity (cf. the dashed linear trend slope as well as the Spearman rank correlation rho). The scatter plots also indicate a temporal dependency of the deviations, as largest deviations tend to relate to events occurring after around 2010, which further shows the importance of the trend representation on the drought response of the products.

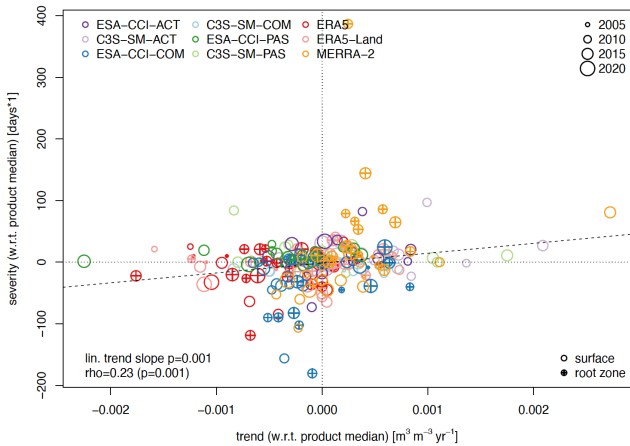
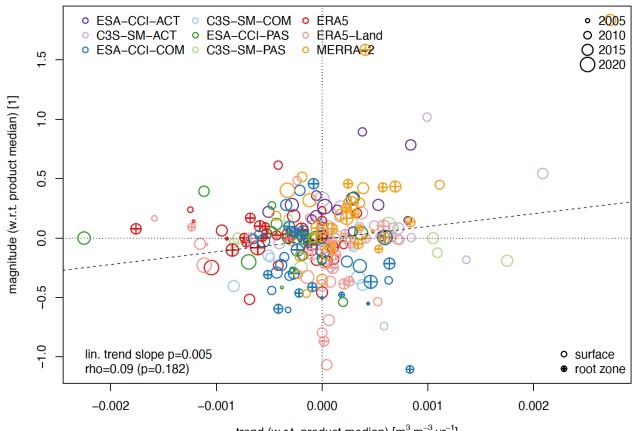

**Figure 12 Product deviations in drought severity (left) and magnitude (right) as a function of product deviations in the 2000–2020 soil moisture trends, with circle sizes depending on the chronology of the events within the investigated period (i.e., later events are displayed with larger circles). Deviations are displayed with respect to the product median of the individual events, separately calculated for the surface and the root zone (the latter additionally indicated with a "+"). The trends and drought metrics are averaged over the respective core of the event regions. The p-values of the linear trend slope (dashed line) and the Spearman rank correlation rho between the drought metrics and the soil moisture trends are noted as well.**

For the ACTIVE products and MERRA-2, the positive deviations in the trends and the reduced drought magnitudes and severities (in case of MERRA-2 mostly due to the root zone) are visible in the location of several events in the respective upper right quadrants. The ACTIVE products also exhibit an intensification and extension of the wetting trends in the later part of the analysis period (compared to the 2000–2020 period, not shown), which further contributes to the reduced drought magnitudes of later events. An additional contribution may come from sensing issues of active microwave remote sensing during dry spells, which lead to an increase in the backscatter of the signal due to subsurface scattering, resulting in an erroneous increase in soil moisture (Wagner et al., 2022).

**5.2 Drivers of soil moisture trends in the reanalysis products**

Thus, the representation of dry-season soil moisture trends of the products is an important factor explaining the differences in their drought responses. To shed light on possible reasons for the observed differences in soil moisture trends between ERA5/ERA5-Land and MERRA-2, Fig. 13 displays the Theil-sen trend estimates on yearly means of dry-season precipitation, runoff, evapotranspiration and 2 m temperature (in addition to the trends in root-zone soil moisture) for these





products as well as for two gridded stations-based datasets of precipitation (GPCC) and temperature (Berkeley Earth). The patterns in precipitation trends (Fig. 13 a–b) appear similar for ERA5 and MERRA-2, except for central Africa (trends mostly negative in ERA5, but mostly positive in MERRA-2), as well as southeast Asia (more widespread positive trends in MERRA-2). These patterns in precipitation trends also mostly agree with the gridded observations (Fig. 13 c), which support the more widespread positive trends in southeast Asia and show a mix of positive and negative trends in central Africa. Pattern correlations with the observed precipitation trends amount to 0.62 for MERRA-2 and to 0.57 for ERA5. Larger differences between the reanalyses can be observed for trends in runoff and particularly for evapotranspiration (Fig. 13 d–i). In large parts of Asia, Africa and north America, trends in evapotranspiration are strongly and widespread positive in MERRA-2 while ERA5/ERA5-Land show more mixed or negative trends in these regions. These differences in evapotranspiration trends are reflected in the described differences in the soil moisture trends (Fig. 13 j–l, cf. Sect. 4.4).

These regional differences in evapotranspiration and soil moisture trends show a clear link to regional differences in 2 m temperature trends (Fig. 13 m–o), which are (more) negative for MERRA-2 in these regions, while ERA5/ERA5-Land there show positive or weaker negative temperature trends. The temperature trends based on gridded observations (Fig. 13 p) in fact agree better with ERA5/ERA5-Land, while MERRA-2 overestimates the negative trends in Asia, Africa, and North America compared to the observations. Corresponding pattern correlations with the observed temperature trends amount to 0.71 for MERRA-2 and to 0.76 for both ERA5 and ERA5-Land, but with a pronounced negative mean bias of $-0.014$ K yr$^{-1}$ of MERRA-2 compared to a smaller positive bias of ERA5 (ERA5-Land) of 0.006 K yr$^{-1}$ (0.009 K yr$^{-1}$).

In contrast to ERA5, MERRA-2 does not benefit from an analysis of synoptic surface air temperature observations (Simmons et al., 2017), and is thus less constrained by ground observations. This could explain the identified negative biases in 2 m temperature trends of MERRA-2 compared to ERA5/ERA5-Land, which includes an assimilation of these ground data. Note that similar patterns of inter-product differences in evapotranspiration, soil moisture and temperature also emerge when considering trends based on the full year instead of dry season only, and when looking at longer-term trends (i.e., 1992–2020 instead of 2000–2020 only; not shown). In both cases, ERA5/ERA5-Land appears more consistent with the observed temperature trends, translating into the observed differences in the evapotranspiration and soil moisture trends.




**Figure 13 Theil-sen trend estimate (2000–2020 period) on yearly means of dry-season precipitation (a–c), runoff (d–f), evapotranspiration (g–i), root-zone soil moisture (j–l) and surface air temperature (m–p) from the reanalysis products as well as two gridded stations-based datasets for precipitation and temperature. Values are in mm/d yr⁻¹ for the fluxes, respectively m³ m⁻³ yr⁻¹ for soil moisture and K yr⁻¹ for temperature. Dry season is defined as 25% climatologically driest days of the year based on ERA5-Land surface soil moisture. Regions with non-significant trends are not masked out for easier comparison with the trends in soil moisture. Note that ERA5-Land is forced by ERA5 precipitation and is thus not shown for the former.**

Overall, the stronger constraint of ERA5/ERA5-Land with observed regional temperature trends seems to result in more

widespread soil drying and evapotranspiration decreases. In contrast, in MERRA-2 the positive trends in precipitation translate into enhanced soil moisture and evapotranspiration and a resulting stronger regional cooling.



Apart from the impact of these differences in the forcing and in the data assimilation strategies on the observed soil moisture trends, differing land-surface model parametrisations may further contribute to product-specific drought representation and

trends of the reanalyses. In particular, MERRA-2 has been shown to exhibit a prolonged surface soil moisture memory (Dirmeyer et al., 2016; cf. Fig. 6 therein), which contributes to the observed partly prolonged durations (and stronger severities) of the investigated droughts in the surface layer of this product. Also, He et al. (2023) report that in water-limited evapotranspiration regimes, MERRA-2 shows a larger overestimation of soil moisture memory times compared to estimates from SMAP, while the bias in ERA5 is lower. It is noted that soil moisture thresholds (i.e., wilting point and critical point;

see e.g., Seneviratne et al., 2010) in the land-surface model parametrisations contribute to the observed differences in the soil moisture memory times. Comparing these soil moisture thresholds between ERA5 (based on HTESSEL) and MERRA-2 (based on the CLSM) reveals that both the wilting point and particularly the critical point tend to be higher for ERA5 than for MERRA-2 (cf. also Schwingshackl et al., 2017, Fig. 14 therein). This may translate into the observed stronger drought representation of ERA5 since it more quickly enters a soil moisture limited evapotranspiration regime during dry downs.

**5.3 Impact of satellite soil moisture retrieval uncertainties and land-surface/bioclimatic variables**

The detected differences in trend (and consequent drought) representation of reanalysis and remote-sensing products (Fig. 11) have partly been reported previously (e.g., Dorigo et al., 2012; Preimesberger et al., 2021). Past studies have linked them to fundamental modelling simplifications in the description of human impacts, which may explain differences regionally (Qiu et al., 2016). However, differences result also from the intrinsic trend representation error of the satellite

products. The evidence of locally contradicting trends between the considered C3S-SM-/ESA-CCI-ACT, -PAS, and -COM (Fig. 14) suggests that the differences in the observation system and retrieval algorithm used in the various products have a non-negligible effect on their trend- and drought-detection capacity.

The presented trend analysis is bound to deal with the heterogeneities in the true spatial support and sampling frequency of

the products, which can explain part of the observed deviations if accounted for explicitly (Wen et al., 2022). In this respect, the data sets are set apart by the lower observational density that results from ingesting (in the reference 2000–2020 period) four sensors in the ACTIVE products against more than double that amount in PASSIVE and COMBINED. This affects the noise levels and – remarkably – their rate of change over time (Hirschi et al., 2023), which leads to biased trends. On top of this, the individual sensors are subject to their own performance drift (Fennig et al., 2020), that propagates to the merged

products but is virtually impossible to isolate in the merged soil moisture signal. Generally, spurious trends are also attributed to non-resolved inter-sensor biases in the merging process (Yang et al., 2013). However, this was not found to be the case in antecedent product versions of the considered C3S and ESA CCI products (Preimesberger et al., 2021; Su et al., 2016).



Dynamic processes on the land-surface present an additional potential impediment to the stability of the soil moisture retrieval (and thus trends and anomalies representation). Retrieval models may in fact be grounded on stationarity assumptions that are challenged by evolving land-surface characteristics. For instance, the vegetation correction of the H SAF ASCAT soil moisture record ingested in the ESA-CCI-/C3S-SM-ACT and -COM products is parametrised on a seasonal basis in the TU Wien model (Naeimi et al., 2009; H SAF, 2021), thus not accounting for inter-annual differences
and trends in vegetation (Vreugdenhil et al., 2016). This may introduce biases over time, leading to an inconsistent representation of the anomalies and to spurious trends in certain areas. The same effect would be caused by temporal variations of the statically calibrated dry- and wet-reference, following soil porosity variations. Abrupt land cover changes are also not automatically parametrised and cause artificial trends which should be visible for instance in areas of widespread deforestation or urban growth (Hahn et al., 2023).


Based on the above, trends in the remote sensing products – or at least differences between them and with the reanalyses – might be explained by considering the underlying trends of relevant land-surface characteristics and bioclimatic indicators. Hence, dry-season trends in soil moisture are compared globally to those calculated using the maximum data availability over the 2000–2020 period for VOD, ERA5-derived aridity (2000–2018 only), fractional inundated area, and fractional
covers of urban area, of bare soil and of tree cover, and the results are included in Appendix B. Where possible (for data at daily resolution, i.e., VOD and inundated area), only the dry-season trends are considered. In case of aridity, there appears to be a strong explanatory capacity for the soil moisture trends in all considered reanalyses – not just ERA5/ERA5-Land, for which this is expected due the model internal consistency – and in most of the satellite-based products (Fig. B1 of the Appendix B). However, the behaviour is not as clear-cut in C3S-SM-ACT and -PAS. Such observation is consistent with the
weaker drought representation found for the ACTIVE and (to a lesser extent) PASSIVE products (Fig. 10). As argued, trends in vegetation cover or density may reflect in the soil moisture signal of the remote sensing products for the role they play in the uncertainty budget, but should also reflect the soil moisture signal as a result of changes in water availability in both satellite and model data. In the case of tree cover (Fig. B2), ESA-CCI-ACT and C3S-SM-COM seem to show a strong relation with (dry) wet trends in areas of (de-)forestation, while ESA-CCI-PAS shows a similarly strong relation, but in the
opposite direction. A positive relation is to a lesser extent also visible in ERA5-Land, ERA5 (root zone only) and ESA-CCI-COM-RZSM (when considering q75 of the trend distributions) as well as in C3S-SM-ACT and MERRA-2 (when considering q25). Conversely, global VOD (Fig. B3) only explains trends in the C3S-SM-PAS and ESA-CCI-ACT products, as well as to a lesser extent in the ERA5/ERA5-Land root zone (considering again q75). This lack of relation in some of the products might be explained by the presence of the hypothesized vegetation parametrisation effect in the retrieval models.
The difference in the two considered ACTIVE data sets indicates that rather than the product type (PASSIVE or ACTIVE), the merging algorithm version or the original sensor record version have an impact on the product trends. The physical relation between VOD and water availability shows more clearly when water limited areas only are considered (below the 25th percentile of the mean ERA5 100–289 cm depth layer soil moisture for the 2000–2020 period, Fig. B4). In this case,





soil moisture emerges more often as a vegetation control in the remote sensing products (Lyons et al., 2021), as captured by
C3S-SM-PAS, and ESA-CCI-ACT/-PAS, and to a smaller extent also by the root-zone of ERA5-Land. Surprisingly, both
ERA5 and MERRA-2 tend to show an opposite behaviour in parts of the distribution. No distinct relations emerge with bare
soil or inundated area trends for either of the products, although a subsurface scattering effect (Wagner et al., 2022) might
explain remarkably wetter trends in the higher bare ground quantiles for both ACTIVE products (Fig. B5). ESA-CCI-ACT
also displays an evident increase in soil moisture trends with trends of urban area fraction, as does C3S-SM-ACT (Fig. B6).
This is consistent with similar observations made for the ASCAT-derived products (Hahn et al., 2023), and visible also in
C3S-SM-/ESA-CCI-COM (which also ingest ASCAT).

In synthesis, several controls can be identified for the soil moisture trends in the various products, although no one variable
explains all trends coherently.  In some cases (e.g., for urban area), these may be artifacts of the L0 signal that should be
decoupled in the retrieval of soil moisture through an update in the model parametrisation. In other cases, it is reasonable to
assume some form of relationship (e.g., for the aridity indicator or for VOD in water-limited regions), which, however, a few
products fail to render.

## 6 Conclusions

We analyse major agroecological droughts of the past two decades with different metrics and test the ability of selected
state-of-the-art long-term reanalysis and remote-sensing soil moisture products to represent these events. The events are
characterised by their severity, magnitude, duration, and spatial extent, which are calculated from standardised daily
anomalies of surface and root-zone soil moisture. We consider well documented drought events selected based on scientific
literature and drought reports and focus on the relative behaviour of the products to circumvent the lack of widely available
ground data of soil moisture. Thus, we do not aim for a in situ validation of the products regarding their representation of the
considered drought events but focus on the product ensemble and identify the products with larger deviations from the
majority to collect convergence of evidence. These resulting product differences in the representation of the drought events
are further placed in the context of trends in dry-season soil moisture and in their drivers.

The investigated products are all able to capture the considered 18 drought events. The ACTIVE microwave remote-sensing
products, and to a lesser extent the PASSIVE products, tend to show weaker drought signals based on surface soil moisture
in all metrics compared to the COMBINED products and ERA5/ERA5-Land. This is most pronounced for the drought
magnitudes of the ACTIVE products. The magnitudes are also reduced in MERRA-2 both in the surface layer and the root
zone. In the root zone (based on the reanalysis products and ESA-CCI-COM-RZSM), the drought events appear dampened
in magnitude and smaller in spatial extent, while a tendency for prolonged durations of the droughts is observable (except for





MERRA-2). ESA-CCI-COM-RZSM displays partly stronger representation of the droughts in all metrics compared to the reanalysis products.

Diverse and partly contradictory global distributions of dry-season soil moisture trends among the products affect the observed product differences in the drought responses. ESA-CCI-COM, ERA5-Land and ERA5 show largest area fractions

of negative dry-season surface soil moisture trends, while MERRA-2, C3S-SM-PAS and -ACT show largest fractions of positive trends. The product deviations in drought severity and magnitude show a relation with their deviations in the dry-season soil moisture trend in the respective event regions. This is most visible in the reduced drought magnitudes of MERRA-2 and the ACTIVE remote sensing products compared to the other products, which is linked to their larger global fractions of strong positive trends in dry-season soil moisture. Also, sensing issues of active microwave remote sensing

during dry spells may contribute to the pronounced weaker drought magnitudes (and wetter trends) of the ACTIVE products. Nevertheless, some regions with largely consistent drying trends are apparent in parts of central Europe, in a region north of the Black Sea/Caspian Sea, in southern Africa, and in parts of Australia, Siberia and South America.

This study demonstrates that soil moisture trends play a fundamental role in the drought representation of different products.

Uncertainties in the representation and global distribution of dry-season soil moisture trends, both from remote sensing and among the reanalysis products, contribute to product-specific representations of droughts, in particular affecting the drought magnitude. The different global patterns of dry-season soil moisture trends of the reanalysis products ERA5/ERA5-Land and MERRA-2 can be related to regional differences in their runoff and particular evapotranspiration trends, while their trends in precipitation are more similar and comparable to observed precipitation trends. The diverse soil moisture and

evapotranspiration trends, however, show a clear link to regional differences in 2 m temperature trends in parts of Asia, Africa, and North America, where MERRA-2 shows a negative bias compared to observed temperature trends. This indicates that ERA5 is more constrained by the observed temperature trends, likely resulting in more realistic drying trends, while MERRA-2 displays widespread wetting trends in these regions.

We also identify several land-surface characteristics and bioclimatic indicators (i.e., aridity, VOD, fractional coverage of urban area, of tree cover and of bare soil) that control soil moisture trends in the various products, although none of these explains all trends coherently. The analysis of trends in these land-surface and bioclimatic variables qualitatively shows that the soil moisture trends are affected by retrieval or modelling artifacts, e.g., due to non-valid stationarity assumptions in the land-surface variables. Conversely, trends in these variables may show valid physical relationships to trends in soil moisture

(e.g., in case of aridity, VOD in water-limited regions), which are however not represented by some products. As a future step, the exact sources of such artifacts should be identified to reconcile the different – and partially diverging – trends representations and advance the drought assessment capacity of the remote sensing observations and reanalysis systems.



## Appendix A

**Table A1** Area fractions (in %) of positive and negative trends within each product. Values in bold are referred to in the manuscript text. Note that trends are not masked for significance, but for common spatial coverage of the datasets.

|  | Dataset | % area positive trends | % area negative trends |
|---|---|---|---|
| **Surface soil moisture** | ESA-CCI-ACT | 49.9 | 48.8 |
|  | ESA-CCI-COM | 29.7 | **70.3** |
|  | ESA-CCI-PAS | 46.9 | 52.9 |
|  | C3S-SM-ACT | **58.9** | 39.8 |
|  | C3S-SM-COM | 44.9 | 55.1 |
|  | C3S-SM-PAS | **59.5** | 40.5 |
|  | ERA5 | 38.5 | **61.5** |
|  | ERA5-Land | 37.8 | **61.6** |
|  | MERRA-2 | **63.6** | 36.4 |
| **Root-zone soil moisture** | ESA-CCI-COM-RZSM | 33.7 | **66.3** |
|  | ERA5 | 37.8 | **62.2** |
|  | ERA5-Land | 42.5 | 57.1 |
|  | MERRA-2 | **64.4** | 35.6 |

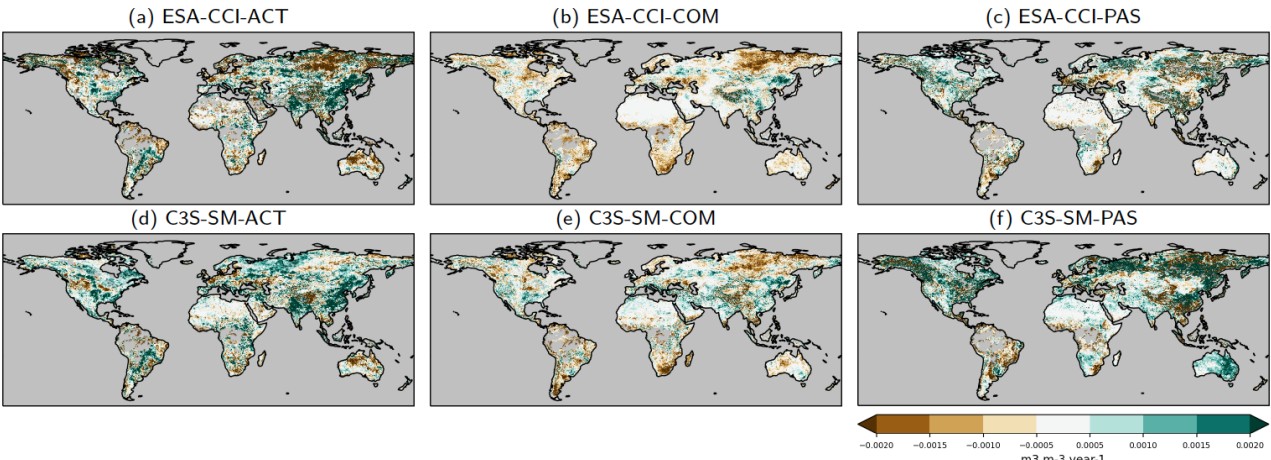

**Figure 14** As Fig. 11, but for surface soil moisture of the different remote sensing products only, including ACTIVE and PASSIVE products. Regions with non-significant trends are not masked out for easier comparison.




**Appendix B**

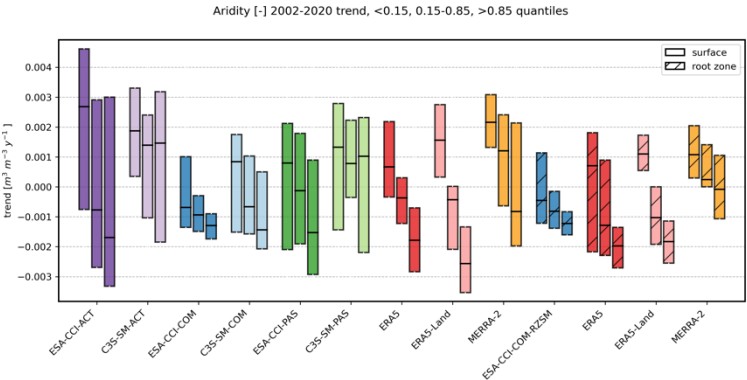

**Figure B1 Distributions (median, inter-quartile range) of global dry-season soil moisture trends in the different products in
relation to different quantile bins of trends in aridity (i.e., <0.15, 0.15-0.85, >0.85). Note that trends are not masked for
significance.**

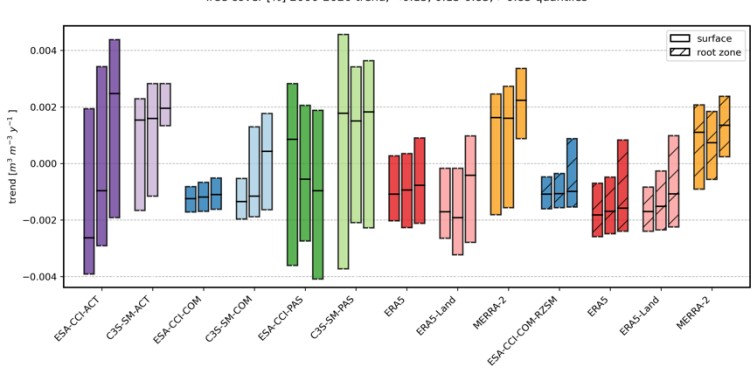

**Figure B2 As Fig. B1, but for trends in tree cover fraction.**


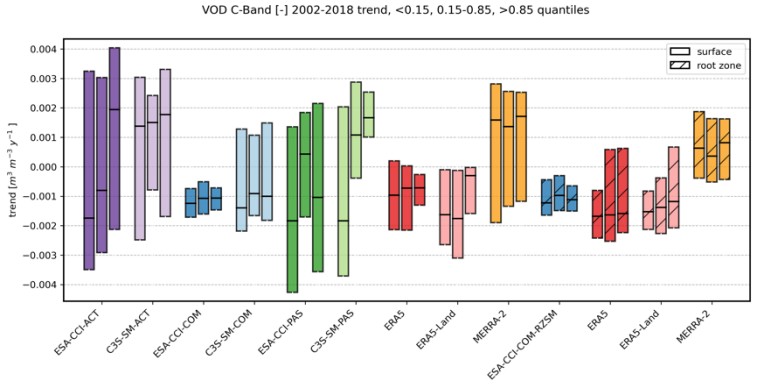

**Figure B3 As Fig. B1, but for trends in global VOD.**



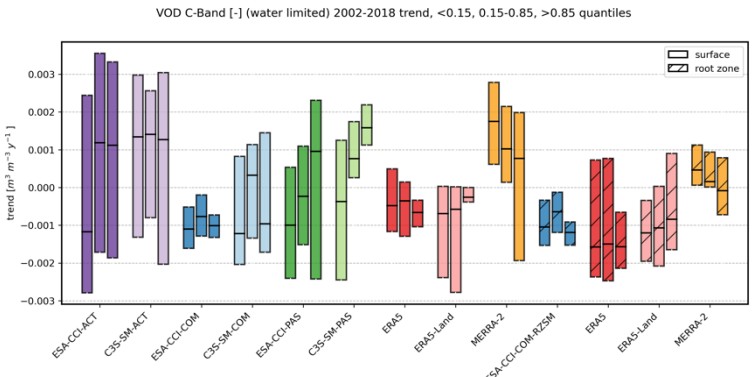

**Figure B4 As Fig. B1, but for trends VOD in water-limited regions.**

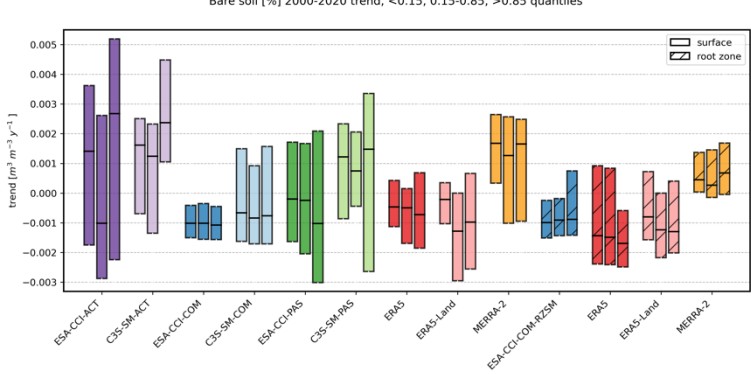

**Figure B5 As Fig. B1, but for trends in bare soil fraction.**

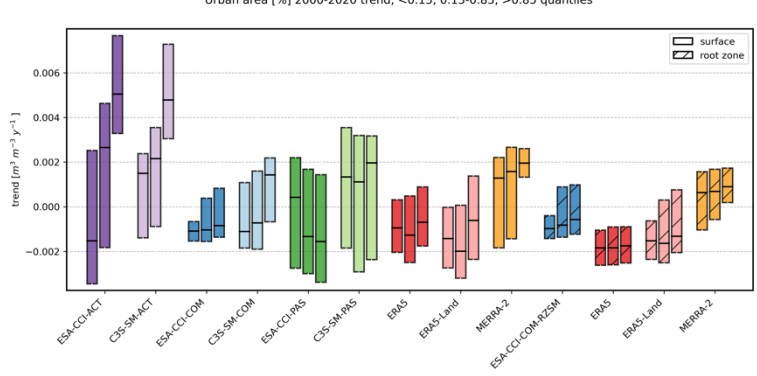


**Figure B6 As Fig. B1, but for trends in urban area fraction.**



**Author contribution**

MH, BC and SIS: conceptualization; MH, BC, PS: Formal analysis, Investigation, Methodology, Visualization; MH and BC: Writing – original draft preparation; all authors: Writing – review & editing.

**Competing interests**

The authors declare that they have no conflict of interest.

**Acknowledgements**

M.H., P.S. and W.D. acknowledge financial support by the ESA's Climate Change Initiative for Soil Moisture (Contract No. 4000126684/19/I-NB). The authors acknowledge the Copernicus Climate Change Service C3S_511 which is being funded by the European Union and Implemented by ECMWF.

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
