# Peer review of "Potential of long-term satellite observations and reanalysis products for characterising soil drying: trends and drought events"

_EGUsphere, 2023_

## Author Comment (AC1)

**Replies to RC1**: Anonymous Referee #1, 18 Dec 2023

This work compares the difference among some soil moisture products in representing the soil moisture drought, and discusses the potential factors that cause this difference. Although the research objective sounds important, the current manuscript is not suggested for publication. The knowledge gap and innovation is not clarified, the implication and suitability of the conclusion is unclear, and the interpretation is confuse and should be revisited carefully. Detailed comments are below:

We thank the reviewer for the critical feedback. Based on this, we decided to reframe the study and focus on the potential of long-term satellite observations for characterising soil drying. Soil drying includes i) long-term negative changes in soil moisture, and ii) agricultural drought events.

Soil moisture trends in long-term satellite observations and differences in these trends between measuring approaches are currently understudied. Most of the available trend analyses use the COMBINED product (e.g., Dorigo et al., 2012; Albergel et al., 2013; Feng and Zhang, 2015; Gu et al., 2019; Preimesberger et al., 2021) and many focus on regional trends only (e.g., Li et al., 2015; Rahmani et al., 2016; Wang et al., 2016; Zheng et al., 2016; An et al., 2016).

However, our analysis shows that soil moisture trends from ACTIVE, PASSIVE and COMBINED products are associated with substantial uncertainties (cf. Fig. 14 and Table A1 of the Appendix A in the current manuscript). Documenting these diverse and partly contradicting trend patterns is crucial to understand where confidence in the remote-sensing products is justified, and where not.

Accordingly, based on the ACTIVE, PASSIVE and COMBINED satellite products, we will identify regions with soil moisture trend direction agreement and those with trend disagreement (products deviate) in order to identify the areas where the agreement leads to higher confidence in satellite observed trends (see Fig. 1, top left).

We will confront this with a similar analysis based on the reanalysis products (Figure 1, top right). Based on the analysis of the drivers of the soil moisture trends in the reanalysis products (cf. Section 5.2, Fig. 13 of the current manuscript), and the relation to observed trends in these drivers (i.e., precipitation, temperature), we will identify the reanalysis products with higher confidence regarding the representation of soil moisture trends.

[Figure]

Figure 1: Trends agreement in the sign (i.e., all agree on positive or negative trends respectively) of the ACTIVE, PASSIVE and COMBINED satellite products (top left), of the ERA5, ERA5-Land and MERRA-2 reanalyses (top right), and of all mentioned products (bottom). Trends are based on surface soil moisture and are not masked for significance.

In a new "Discussion and synthesis" section, we will provide a synthesis of the global trends based on the "best-estimate" products from both remote sensing and reanalysis data. This synthesis will consider the analysis of the areas with trend agreement (cf. Fig. 1) and also make use of the adjusted area fractions of positive and negative trends of the products (cf. Table A1 of the current manuscript).

In a second step, we will investigate the agricultural drought events as a use case to quantify how the diverse trend representation also affects the drought detection capabilities of the products – evidence in this sense is already present in the current manuscript and will be made more prominent. For this, we will stratify the product intercomparison of the drought metrics (e.g., magnitude, severity), particularly regarding the relation of product deviations in drought representation and soil moisture trends, by separating the drought regions in areas with trends agreement and in those without agreement. This will allow to generalise the product intercomparison.

1. The innovation. The introduction states the importance of the drought and then states that "involved products show partly considerable differences in the global patterns and magnitudes of the soil moisture drying.". However, either a comprehensive review on the literature that evaluates the ability of different products in capturing drought, or the current knowledge gap on understanding the differences between different products, is provided. This makes it confuse to the reader on the innovation of the current work.

   As stated above, soil moisture trends in long-term satellite observations and differences in these trends between measuring approaches are currently

understudied. Most of the available trend analyses use the COMBINED product and many focus on regional trends only (cf. references above). The COMBINED product, however, is based on the merging of the individual ACTIVE and PASSIVE sensors.

Thus trend disagreement in these underlying products and with the merged product is a clear indication of problems in the data (addressed in Section 5.3 of the current manuscript), and these may also translate into the COMBINED product. On the other hand, trend agreement may indicate regions where confidence in the remote-sensing products is justified. The output of the study is critical feedback on the products to prompt an investigation and reconciliation of (the causes of) such trends in the upcoming versions.

We will extend the literature review on currently available trend assessments to more clearly indicate the innovation of the study.

2. The implication and suitability of the conclusion. The current result is based on the intercomparison between different datasets based on a few drought cases (e.g., 19), so the results only indicate the difference between the chosen products (e.g., ESA-CCI, ERA5, ERA5_Land and MERRA2). Then, what is the implication of the results? Which dataset should we relief on? Or which dataset is more suitable to perform drought analysis? In addition, the drought cases are mainly over the Europe and are not enough for a global perspective.

We will reframe the study by first investigating the global soil moisture trends based on the considered products and then looking at the impact of the diverse trend patterns on their drought detections capabilities. In a new "Discussion and synthesis" section, we will provide a synthesis of the global soil moisture trends based on the best-estimate products.

Based on the analysis of the drivers of the soil moisture trends in the reanalysis products (cf. Section 5.2 of the current manuscript), there exists an indication to favour ERA5/ERA5-Land over MERRA-2 when taking into account the negative temperature bias of the latter as discussed in the manuscript. Also, since the drying patterns of the COMBINED products tend to agree more closely with ERA5/ERA5-Land, to favour it over the ACTIVE and PASSIVE products. This is also due to the discussed artefacts of the ACTIVE products in urban areas and its sub-surface scattering effects.

While the goal is not to provide a definitive indication of a single product to use for trends assessment, a substantiated and reliable indication of regions of confidence will be provided. We will be more clear on this in a revised version and we will use these findings to provide a synthesis of the global soil moisture trends based on the best-estimate products from both remote sensing and reanalysis data. This will be presented in a new "Discussion and synthesis" section and will consider the analysis of the trend agreement areas (cf. Fig. 1 above), as well as the area fractions of positive and negative trends (cf. Table A1 of the current manuscript).

We will further investigate seasonal drought events as a use case to show how the diverse trend representation also affects the drought detection capabilities of the

products. For this, we will stratify the product intercomparison of the drought metrics (e.g., magnitude, severity), particularly regarding the relation of product deviations in drought representation and soil moisture trends, by separating the drought regions in areas with trends agreement and in those without agreement. This will allow to generalise the product intercomparison.

3.  The dry-season SM. The dry-season SM in current research is discontinuous, and is different from the usually used concept that is based on a consecutive period with lower SM. Therefore, the meaning of the the linear trend of dry-season SM should be clarified more clearly. In addition, the trend of dry-season SM is used to interpret the difference among different products in representing drought characteristics. This is very confuse to me, because lots of the drought cases happened during the wet seasons (e.g., June-September).

    We agree with the reviewer that some of the events will not be fully covered by the dry season. In order to circumvent this we decided to switch to trends based on the full year, but excluding the frost period in this case (see below). Previous analyses show that trend patterns based on the full year (Fig. 2) are comparable to dry-season only trend patterns.

[Figure]

    Figure 2: 2000-2021 trends of ESA CCI v08.1 COMBINED, ACTIVE and PASSIVE. Trends are based on the full year, and a Mann-Kendall test with a false rejection rate of 0.05 was performed to mask out regions where no significant trend is present. Figure taken from Hirschi et al., 2023.

4.  The different spatial resolution of products. Was the analysis based on the original spatial resolution of different datasets or a fixed resolution (e.g., aggravate them to 0.25°)? Different spatial resolution would lead to different grid samples in the same drought area, and may influence the result. In addition, the high-resolution products tend to be more heterogeneous and potentially influence the identification of the core zones of drought events.

    The current analysis is based on the original resolution of the products with the idea to also consider the added value of the higher spatial resolution of ERA5-Land with 0.1° vs. ERA5 with 0.25°. Using ERA5-Land resampled to 0.25° instead of 0.1° had only minor effects on the trend patterns and the drought representation.

    However, we will consider switching to a fixed resolution of 0.5° in a revised manuscript to simplify the product intercomparison and the consistent soil frost masking.

5. It seems that, the soil moisture in reanalysis products includes both liquid and solid soil water while the remote sensing products only provide the liquid soil water. I suggest the author to confirm this and pay attention to the frozen period when comparing different products.

We agree on this fact, even though the analysed events typically do not fall within the soil frost period. However, given reviewer's point 3 on the dry season, we decided to switch to trends based on the full year, but in this case excluding the soil frost period.

We will apply a frozen soil mask based on the individual soil temperature data for the reanalysis products, and then apply a mutual masking of all products (note that the remote sensing products are already masked for frozen soil conditions).

6. The discussion said that satellite datasets do not consider the dynamic land-surface characteristics and bioclimati and attributes the differences between satellitedataset and reanalyses dataset to the considering of the underlying trends of relevant land-surface characteristics and bioclimatic indicators. However, similar with the satellite dataset, the reanalysis dataset also does not consider these dynamic factors. Therefore, the discussion may be incorrect.

We agree with the reviewer that both remote sensing and reanalysis products do not directly consider the temporal dynamics of land-surface characteristics (and bioclimatic indicators). We will also note this in the discussion for the reanalysis products. However, unlike the remote sensing products, the reanalyses assimilate a variety of ground data that are at least indirectly affected by potential changes in the land-surface properties.

References:
Albergel, C., Dorigo, W., Reichle, R. H., Balsamo, G., de Rosnay, P., Munoz-Sabater, J., Isaksen, L., de Jeu, R., and Wagner, W.: Skill and Global Trend Analysis of Soil Moisture from Reanalyses and Microwave Remote Sensing, Journal of Hydrometeorology, 14, 1259-1277, doi:10.1175/jhm-d-12-0161.1, 2013.

An, R., Zhang, L., Wang, Z., Quaye-Ballard, J. A., You, J. J., Shen, X. J., Gao, W., Huang, L. J., Zhao, Y. H., and Ke, Z. Y.: Validation of the ESA CCI soil moisture product in China, Int J Appl Earth Obs, 48, 28-36, doi:10.1016/j.jag.2015.09.009, 2016.

Dorigo, W., de Jeu, R., Chung, D., Parinussa, R., Liu, Y., Wagner, W., and Fernandez-Prieto, D.: Evaluating global trends (1988-2010) in harmonized multi-satellite surface soil moisture, Geophysical Research Letters, 39, doi:10.1029/2012gl052988, 2012.

Feng, H. and Zhang, M.: Global land moisture trends: drier in dry and wetter in wet over land, Sci Rep, 5, 18018, doi:10.1038/srep18018, 2015.

Gu, X. H., Li, J. F., Chen, Y. D., Kong, D. D., and Liu, J. Y.: Consistency and Discrepancy of Global Surface Soil Moisture Changes From Multiple Model-Based Data Sets Against Satellite Observations, Journal of Geophysical Research-Atmospheres, 124, 1474-1495, doi:10.1029/2018jd029304, 2019.

Hirschi, M., Stradiotti, P., Preimesberger, W., Dorigo, W., and Kidd, R.: Product Validation and Intercomparison Report (PVIR): Supporting Product version v08.1. Deliverable D4.1 Version 1, ESA Climate Change Initiative Plus - Soil Moisture, doi:10.5281/zenodo.8320930, 2023.

Li, X. W., Gao, X. Z., Wang, J. K., and Guoa, H. D.: Microwave soil moisture dynamics and response to climate change in Central Asia and Xinjiang Province, China, over the last 30 years, J Appl Remote Sens, 9, doi:10.1117/1.Jrs.9.096012, 2015.

Preimesberger, W., Scanlon, T., Su, C.-H., Gruber, A., and Dorigo, W.: Homogenization of Structural Breaks in the Global ESA CCI Soil Moisture Multisatellite Climate Data Record, IEEE Transactions on Geoscience and Remote Sensing, 59, 2845-2862, doi:10.1109/tgrs.2020.3012896, 2021.

Rahmani, A., Golian, S., and Brocca, L.: Multiyear monitoring of soil moisture over Iran through satellite and reanalysis soil moisture products, Int J Appl Earth Obs, 48, 85-95, doi:10.1016/j.jag.2015.06.009, 2016.

Wang, S. S., Mo, X. G., Liu, S. X., Lin, Z. H., and Hu, S.: Validation and trend analysis of ECV soil moisture data on cropland in North China Plain during 1981-2010, Int J Appl Earth Obs, 48, 110-121, doi:10.1016/j.jag.2015.10.010, 2016.

Zheng, X. M., Zhao, K., Ding, Y. L., Jiang, T., Zhang, S. Y., and Jin, M. J.: The spatiotemporal patterns of surface soil moisture in Northeast China based on remote sensing products, J Water Clim Change, 7, 708-720, doi:10.2166/wcc.2016.106, 2016.

---

## Author Comment (AC2)

**Replies to RC2**: Anonymous Referee #2, 28 Dec 2023

This study investigates the ability of surface and root-zone soil moisture from multiple reanalysis and remote-sensing products in representing drought events in recent 20 years globally, and compares their differences in describing various drought metrics. Overall, this paper provides a comprehensive reference for selecting datasets for drought study. But the structure and conclusions of this article are not clear enough for including too many datasets and drought events, so I suggest a major revision before publication. The main suggestions are as follows.

General comments:

The authors should be more familiar to Europe, and nearly half of the 18 selected events occurred over Europe. So why not just focus on the ability of multiple datasets in characterising seasonal drought events in Europe? In Figures 6－7 and 10, the drought metrics show remarkably discrepancies between seasonal and multi-year events. Thus I suggest the reconsideration of the clarification.

We thank the reviewer for the valuable feedback. Based on the comments of Reviewer #1, we decided to reframe the study and focus on the potential of long-term satellite observations for characterising soil drying. This includes i) long-term negative changes in soil moisture, and ii) agricultural drought events.

Thus we will first focus on the global soil moisture trends, which will be based on the full year instead of dry season only. Using the ACTIVE, PASSIVE and COMBINED satellite products, we will identify regions with soil moisture trend direction agreement and those with trend disagreement (products deviate) in order to identify the areas where the agreement leads to higher confidence in satellite observed trends. We will confront this with a similar analysis based on the reanalysis products.

In a new "Discussion and synthesis" section, we will then provide a synthesis of the global trends based on the "best-estimate" products from both remote sensing and reanalysis data. This synthesis will be based on the analysis of the areas with trend agreement and disagreement and will consider the adjusted area fractions of positive and negative trends (cf. Table A1 of the current manuscript).

We will further investigate seasonal drought events as a use case to show how the diverse trend representation also affects the drought detection capabilities of the products. For this, the product intercomparison of the drought metrics (e.g., magnitude, severity), particularly regarding the relation of product deviations in these metrics and soil moisture trends, will be stratified by separating the drought regions in areas with trends agreement and in those without agreement. This stratified analysis based on the trend agreement will allow to generalise the product intercomparison. We will consider seasonal events only in the drought analysis (and neglect the few multi-year events) in order to not overload the paper and to allow better comparability of the events.

Specific comments:

1. The description of data and methods (section 2 and 3) are too long. Although the detailed information may be helpful to readers, it is not suitable in a scientific paper.

   We will shorten the description of the datasets. In particular, we will not consider the C3S soil moisture product anymore, since it is based on a precursor version of the processing algorithm of ESA CCI and thus does not represent the latest product achievements of merged satellite products. Also, as a suggestion from Reviewer #3, referenced literature on the validation of the products will be moved and only considered in the discussion section to better link the findings of the analysis.

2. The figures and tables are not well organized in the paper structure. The quantitative results in tables can be integrated to the respective figures, which can make it more clear and comparable to readers. For example, the area mean of severity, magnitude and duration in Table 2 can be added to Figure 1－3, and the maximum of spatial extent of the events to Figure 5. In addition, Figure 4－5 can also be integrated in a Figure as (a) and (b), respectively.

   We thank the reviewer for the detailed suggestions on the organisation of the figures and tables. We agree that the numbers of Table 2 can be integrated into the corresponding Figures 1–3 and will adjust the manuscript accordingly. We will also combine Figure 4 and 5 as suggested.

3. In term of the evaluation for the selected drought events, more statistical metrics can be included, such as pattern correlation, RMSE, and so on. Figures 6－9 are displayed only in bars, which is not concise and explicit enough. I recommend the Table graphic type to present each evaluation result for all events and all datasets. The detailed procedure can be seen at https://www.ncl.ucar.edu/Applications/table.shtml.

   Indeed, the presentation of the drought response as barplots may be overwhelming. We will consider the proposed presentation of these results in a revised manuscript.

4. The analysis of dry-season soil moisture is less related with the research objective. I think it is more reasonable to further compare the soil moisture during drought events after presenting the results for multiple drought events.

   As indicated in the replies to Reviewer #1, the dry-season trends will no longer be used and we will refocus the study on soil moisture trends based on the full year (but excluding the soil frost period).

5. As for the long-term trend, the analysis may be better to be conducted for the drought events rather than another indicator.

   We do not think that trends based on the events are meaningful in this case since the events are scattered in space and time. But as mentioned, we will restructure the analysis and first focus on the global soil moisture trends.

The discussion section is not convincing and substantial. In 5.1, For drought metrics and dry-season SM trend were derived from the same variable, they must be related. In 5.2, the attribution method is too simple and no quantitative results are shown.

By reframing the study and focussing on the potential of long-term satellite observations for characterising soil drying, we will investigate seasonal drought events as a use case to show the impact of the diverse trend representations on the drought detection capabilities of the products. As mentioned, the product intercomparison of the drought metrics (e.g., magnitude, severity) will be stratified by separating the drought regions in areas with trends agreement and in those without agreement. Hence, the aim will be to quantify the impact of the trend-drought relation rather than point to its existence. Furthermore, this stratified analysis based on the trend agreement will allow to generalise the product intercomparison.

As for 5.2, we will add statistical metrics (e.g., pattern correlations between the different variables) to better attribute the differences in soil moisture trends to the driving variables.

---

## Author Comment (AC3)

**Replies to RC3**: Anonymous Referee #3, 28 Dec 2023

The study investigates the ability of active and passive based remote sensing soil moisture products and land reanalyses to capture documented drought events and drought trends during the period 2000-2020. The drought events are characterised in different parts of the world by their severity, duration and spatial extent. The events are placed in the context of dry season soil moisture trends and potential reasons for diverging soil moisture trends between the different products are investigated. It is found that all the products capture the selected drought events. Significant differences between the products are found – for example, responses in surface soil moisture tend to be weakest for the active remote sensing products. For the global reanalyses, ERA5 and ERA5-land have a greater tendency for drying trends, whilst MERRA-2 has a greater tendency for wetting trends. Based on other reanalysis variables (evapotranspiration, runoff, precipitation) and observational data, it would appear that the ERA5 and ERA5-land trends are more reliable overall.

The authors have done a detailed and robust evaluation of the different products and have done well to disentangle the reasons (or potential reasons) for the divergences in the results. However, I think the introduction and discussion sections need to be more concise, with some of the detail removed. Further, I think the paper could be strengthened by linking the results to studies where reference soil moisture datasets (e.g. in situ data) have been used to validate drought events and trends (e.g. Li et al., 2020). This would give more weight to the conclusions of the study. Furthermore, I think the rationale for the approach used in this study needs to be more clearly communicated in the abstract and conclusion. Please also see the minor comments below.

We thank the reviewer for the positive feedback.

As a response to Reviewer #1, we decided to reframe the study and focus on the potential of long-term satellite observations for characterising soil drying, which includes i) long-term negative changes in soil moisture, and ii) agricultural drought events. Thus we will first focus on the global soil moisture trends, and in a second step investigate the agricultural drought events as a use case to show the impact of the diverse trend representation on the drought detection capabilities of the products. This change will also be reflected in the introduction and the discussion sections.

To strengthen the link with existing literature, we will move the referenced literature on the validation of the products, which are currently cited in the dataset section, to the discussion and extend it. This will also help to shorten this Section 2 as requested by Reviewer #2.

Line 46: Replace "trigger" with "triggers"

Will be replaced if still applicable in the restructured manuscript.

Lines 76-94: I agree with the rationale for the evaluation approach. However, I still think the authors should link the results to studies where reference datasets have been used in the discussion section (5), as it would reinforce the findings in this study.

We will move (and extend) the referenced literature on validation of the products in the discussion.

Line 148: Suggest to replace "as for" with "to"

The description of C3S soil moisture will be removed since the product will not be considered anymore in a revised manuscript (cf. reply to Reviewer #2).

Section 2.1.2 ERA5

It is important to mention here that in ERA5, T2m/RH2m pseudo observations are assimilated in the soil moisture analysis (see for example de Rosnay et al., 2013). These observations tend to have an important impact on root-zone soil moisture and latent/sensible heat fluxes with the atmosphere (see e.g. Fairbairn et al., 2019). The sensitivity of ERA5 to drought events could potentially be increased by assimilating these observations.

Thanks for this additional information. We will include it in the description of ERA5.

Line 381: Suggest to replace "the average" with "average"

Will be replaced.

Line 399: Suggest to replace "of" with "for"

Will be replaced.

Line 401-403: Please rephrase for clarity and maybe split into two sentences.

Will be rephrased.

Line 497: Suggest to replace "and display" with "display"

Will be changed accordingly.

Line 526: Sentence starting with "Despite the considerable spread…" Please rephrase as sentence does not make sense.

The sentence will be rephrased.

Line 532: Suggest to replace "largest deviations" with "the largest deviations"

Will be changed accordingly.

Line 564: Sentence starting with "These regional differences…". Please rephrase for clarity.

We will rephrase this sentence.

Line 570: Suggest to replace "of MERRA-2" with "for MERRA-2" and replace "of ERA5" with "for ERA5".

Will be changed accordingly.

De Rosnay, P., Drusch, M., Vasiljevic, D., Balsamo, G., Albergel, C. and Isaksen, L., 2013. A simplified extended Kalman filter for the global operational soil moisture analysis at ECMWF. *Quarterly Journal of the Royal Meteorological Society*, *139*(674), pp.1199-1213.

Fairbairn, D., de Rosnay, P. and Browne, P.A., 2019. The new stand-alone surface analysis at ECMWF: Implications for land–atmosphere DA coupling. *Journal of Hydrometeorology*, *20*(10), pp.2023-2042.

---

## Author Response (AR1)

**General comment to editorial office:**

Note that we have revised the colour schemes for denoting the individual products as a request from the preceding submission.

**Replies to RC1**: Anonymous Referee #1, 18 Dec 2023

This work compares the difference among some soil moisture products in representing the soil moisture drought, and discusses the potential factors that cause this difference. Although the research objective sounds important, the current manuscript is not suggested for publication. The knowledge gap and innovation is not clarified, the implication and suitability of the conclusion is unclear, and the interpretation is confuse and should be revisited carefully. Detailed comments are below:

We thank the reviewer for the critical feedback. Please find in the following our replies (in blue), together with pasted actual changes from the manuscript in selected cases (in order not to overload this document). Else, we refer to the respective sections of the revised manuscript.

Based on the feedback, we decided to reframe the study and focus on the potential of long-term satellite observations and selected reanalysis products for characterising soil drying. Soil drying includes i) long-term negative changes in soil moisture, and ii) agricultural drought events. This change in the focus is also reflected in a change of the manuscript structure as well as its title, which now reads: "Potential of long-term satellite observations and reanalysis products for characterising soil drying: trends and drought events".

Soil moisture trends in long-term satellite observations and differences in these trends between measuring approaches are currently understudied. Most of the available trend analyses use the COMBINED product (e.g., Dorigo et al., 2012; Albergel et al., 2013; Feng and Zhang, 2015; Gu et al., 2019; Preimesberger et al., 2021) and many focus on regional trends only (e.g., Li et al., 2015; Rahmani et al., 2016; Wang et al., 2016; Zheng et al., 2016; An et al., 2016).

However, our analysis shows that soil moisture trends from ACTIVE, PASSIVE and COMBINED products are associated with substantial uncertainties (cf. Fig. 1 and Table 2 of the revised manuscript). Documenting these diverse and partly contradicting trend patterns is crucial to understand where confidence in the remote-sensing products is justified, and where not.

Accordingly, based on the ACTIVE, PASSIVE and COMBINED satellite products, we identify regions with soil moisture trend direction agreement and those with trend disagreement (products deviate) in order to identify the areas where the agreement leads to higher confidence in satellite observed trends (see Supplementary Fig. 1a, also included below).

We confront this with a similar analysis based on the reanalysis products (Supplementary Fig. 1b). Based on the analysis of the drivers of the soil moisture trends in the reanalysis products (cf. Section 4.2, Fig. 2 of the revised manuscript), and the relation to observed trends in these drivers (i.e., precipitation, temperature), we identify the reanalysis products

with higher confidence regarding the representation of soil moisture trends. This is accompanied by a discussion of studies that validate the considered products using reference soil moisture data to link our results to previous work (new section "5.1 Synthesis on soil moisture trends"). The discussed literature supports our findings on the biases in MERRA-2 (cf. Section 4.2) but also ESA-CCI-ACT soil moisture trends.

[Figure]

**Supplementary Figure 1 Agreement in the trend direction of surface soil moisture within (a) the ESA CCI soil moisture products, (b) the ERA5, ERA5-Land and MERRA-2 reanalyses, and (c) all considered remote-sensing and reanalysis products. White colour denotes areas with no consensus in trend direction, green areas with wetting trend agreement, and brown areas with drying trend agreement. Trends are not masked for significance.**

In Section 5.1, we then further provide a synthesis of the global soil moisture trends based on the "best-estimate" products from both remote sensing and reanalysis data. This synthesis considers the analysis of the areas with trend direction agreement (cf. Fig. 9 of the revised manuscript) and makes use of the area fractions of positive and negative trends of the products (cf. Table 2 of the revised manuscript).

In a second step, we investigate the agricultural drought events as a use case to investigate how the diverse trend representation also affects the drought-detection capacity of the products. For this, we stratify the analysis on the relation of product deviations in drought representation and soil moisture trends by separating the drought regions in areas with drying trends agreement and in those without trend direction agreement.

1. The innovation. The introduction states the importance of the drought and then states that "involved products show partly considerable differences in the global patterns and magnitudes of the soil moisture drying.". However, either a comprehensive review on the literature that evaluates the ability of different products in capturing drought, or the current knowledge gap on understanding the differences between different products, is provided. This makes it confuse to the reader on the innovation of the current work.

   As stated above, soil moisture trends in long-term satellite observations and differences in these trends between measuring approaches are currently understudied. Most of the

available trend analyses use the COMBINED product and many focus on regional trends only (cf. references above). The COMBINED product, however, is based on the merging of the individual ACTIVE and PASSIVE sensors.

Thus, trend disagreement in these underlying products and with the merged product is a clear indication of problems in the data, and these may also translate into the COMBINED product. On the other hand, trend agreement may indicate regions where confidence in the remote-sensing products is justified. The output of the study is critical feedback on the products to prompt an investigation and reconciliation of (the causes of) such trends in the upcoming versions. Also, the impact of these uncertainties in the soil drying on the representation of droughts is understudied.

We now more clearly highlight this knowledge gap and the innovation of the study in the Introduction by extending the literature review on currently available trend assessments.

Since in situ observations of soil moisture are still scarce and not continuously available in space and time over long time periods (Dorigo et al., 2011; Dorigo et al., 2021b), reanalysis and merged remote-sensing products provide an alternative for global long-term timeseries to investigate drying trends and soil moisture droughts on supra-regional scales. Here, we investigate the ability of the long-term remote-sensing dataset ESA CCI soil moisture (encompassing multi-sensor merged ACTIVE, PASSIVE and COMBINED surface soil moisture products, as well as a new root-zone soil moisture product based on COMBINED) and selected state-of-the-art reanalysis products (ERA5, the offline ERA5-Land, and MERRA-2) for characterising soil drying. Soil moisture trends in long-term satellite observations and differences in these trends between measuring approaches are still understudied. Most of the available ESA CCI soil moisture based trend analyses use the COMBINED product (e.g., Dorigo et al., 2012; Albergel et al., 2013; Feng and Zhang, 2015; Gu et al., 2019; Preimesberger et al., 2021) and many focus on regional trends only (e.g., Li et al., 2015; Rahmani et al., 2016; Wang et al., 2016; Zheng et al., 2016; An et al., 2016; Jia et al., 2018). Previous analyses indicated that global trend patterns of ESA CCI COMBINED soil moisture may be subject to differences between product versions (Hirschi et al., 2023), due to yet unknown reasons. Even though the patterns have become more stable with latest product versions, potential sources of uncertainty include the different merging steps involved in the ESA CCI processing chain, changes in the sensor composition between versions, and characteristics of the underlying ACTIVE and PASSIVE products per se that translate into the COMBINED product. Understanding where confidence in the remote-sensing based soil moisture trends is justified, and where not, is thus fundamental for the use of such products as climate data record. The same applies to (land) reanalysis products, which we use as a comparison. To attribute some of the product differences, potential drivers of the global soil moisture trends in the reanalyses are analysed by considering trends in relevant variables of the land water balance and surface air temperature, and corresponding trends in ground observational data. Additionally, we look at bioclimatic indicators and land-surface characteristics that potentially affect the stability of the soil moisture retrieval and the reanalysis-based soil moisture.

2. The implication and suitability of the conclusion. The current result is based on the intercomparison between different datasets based on a few drought cases (e.g., 19), so the results only indicate the difference between the chosen products (e.g., ESA-CCI, ERA5, ERA5_Land and MERRA2). Then, what is the implication of the results? Which dataset should we relief on? Or which dataset is more suitable to perform drought analysis? In addition, the drought cases are mainly over the Europe and are not enough for a global perspective.

We reframed the study by first investigating the global soil moisture trends based on the considered products (Section 4.1) and then looking at the impact of the diverse trend

patterns on their drought-detection capacity. In a new section "5.1 Synthesis on soil moisture trends", we provide a synthesis of the global soil moisture trends based on the best-estimate products.

Based on the analysis of the drivers of the soil moisture trends in the reanalysis products (cf. Section 4.2 of the revised manuscript), there exists a clear indication to favour ERA5/ERA5-Land over MERRA-2 when taking into account the positive bias in precipitation trends of the latter and its larger regional deviations from observed temperature trends as discussed in the manuscript. Also, since the drying patterns of the COMBINED products tend to agree more closely with ERA5/ERA5-Land, to favour it over the ACTIVE and PASSIVE products. This is also due to the discussed artefacts of the ACTIVE products in urban areas and its sub-surface scattering effects.

While the goal is not to provide a definitive indication of a single product to use for trends assessment, a substantiated and reliable indication of regions of confidence is provided. We are more explicit on this in the revised manuscript, and we use these findings to provide a synthesis of the global soil moisture trends based on the best-estimate products from both remote sensing and reanalysis data. This is presented in a new section "5.1 Synthesis on soil moisture trends" and considers the analysis of the areas with trend direction agreement (cf. Fig. 9 of the revised manuscript, also included below), as well as the area fractions of positive and negative trends (cf. Table 2 of the revised manuscript).

[Figure]

**Figure 9 Best-estimate products average of 2000–2022 Theil-Sen trends (m³ m⁻³ yr⁻¹) on yearly mean (a) surface and (b) root-zone soil moisture. Underlying daily data is masked based on the ESA-CCI-COM (a) (and -RZSM, b) data availability, and non-frozen soil conditions of ERA5-Land (a, b). The mean trends are only shown in areas with trend direction agreement of both respective best-estimate products, while white colour denotes no consensus in the trend direction. Additionally, areas of common significant trends are hatched.**

We further investigate seasonal drought events as a use case to show how the diverse trend representation also affects the drought-detection capacity of the products (Section 4.3 – 4.5). For this, we stratify the product intercomparison of the drought metrics, particularly regarding the relation of product deviations in the drought magnitude and the soil moisture trends, by separating the drought regions in areas with drying trends agreement and in those without trend direction agreement (Section 5.2). This stratified analysis based on the trend agreement allows to generalise the analysis of the impact of the trend representation on the products' drought-assessment capacity.

3. The dry-season SM. The dry-season SM in current research is discontinuous, and is different from the usually used concept that is based on a consecutive period with lower SM. Therefore, the meaning of the the linear trend of dry-season SM should be clarified

more clearly. In addition, the trend of dry-season SM is used to interpret the difference among different products in representing drought characteristics. This is very confuse to me, because lots of the drought cases happened during the wet seasons (e.g., June-September).

We agree with the reviewer that some of the events may not have been fully covered by the dry season. To circumvent this, we decided to switch to trends based on the full year but excluding the frost period in this case (see below). Previous analyses showed that trend patterns based on the full year (e.g., Hirschi et al., 2023; based on monthly data) are comparable to dry-season only trend patterns. This is confirmed by our new analysis (cf. Fig. 1 of the revised manuscript). Thus, the overall conclusions e.g., regarding the differing fractions of positive and negative trends among the products remain robust with this change.

4.  The different spatial resolution of products. Was the analysis based on the original spatial resolution of different datasets or a fixed resolution (e.g., aggravate them to 0.25°)? Different spatial resolution would lead to different grid samples in the same drought area, and may influence the result. In addition, the high-resolution products tend to be more heterogeneous and potentially influence the identification of the core zones of drought events.

The original analysis was based on the original resolution of the products with the idea to also consider the added value of the higher spatial resolution of ERA5-Land with 0.1° vs. ERA5 with 0.25°. Using ERA5-Land resampled to 0.25° instead of 0.1° had only minor effects on the trend patterns and the drought representation.

However, we switched to a fixed resolution of 0.5° in the revised manuscript to simplify the product intercomparison and the consistent soil frost masking (see Section 2.1).

5.  It seems that, the soil moisture in reanalysis products includes both liquid and solid soil water while the remote sensing products only provide the liquid soil water. I suggest the author to confirm this and pay attention to the frozen period when comparing different products.

We agree on this fact. Given reviewer's point 3 on the dry season, we decided to switch to trends based on the full year in the revised manuscript, but in this case excluding the soil frost period (see Section 3.1 of the revised manuscript).

For this, we apply a frozen soil mask based on the individual soil temperature data for the reanalysis products, and then apply a mutual masking of all products (note that the ESA CCI remote sensing products are already masked for frozen soil conditions).

6.  The discussion said that satellite datasets do not consider the dynamic land-surface characteristics and bioclimate and attributes the differences between satellite dataset and reanalyses dataset to the considering of the underlying trends of relevant land-surface characteristics and bioclimatic indicators. However, similar with the satellite dataset, the reanalysis dataset also does not consider these dynamic factors. Therefore, the discussion may be incorrect.

We agree with the reviewer that both remote sensing and reanalysis products do not directly consider the temporal dynamics of land-surface characteristics (and bioclimatic indicators), and in fact the Section 5.3 already considered the reanalysis products in the figures. We now note this more clearly in the discussion for the reanalysis products and changed the title of Section 5.3 to "Impact of land-surface/bioclimatic variables on satellite soil moisture retrieval and modelling uncertainties".

References:
Albergel, C., Dorigo, W., Reichle, R. H., Balsamo, G., de Rosnay, P., Munoz-Sabater, J., Isaksen, L., de Jeu, R., and Wagner, W.: Skill and Global Trend Analysis of Soil Moisture from Reanalyses and Microwave Remote Sensing, Journal of Hydrometeorology, 14, 1259-1277, doi:10.1175/jhm-d-12-0161.1, 2013.

An, R., Zhang, L., Wang, Z., Quaye-Ballard, J. A., You, J. J., Shen, X. J., Gao, W., Huang, L. J., Zhao, Y. H., and Ke, Z. Y.: Validation of the ESA CCI soil moisture product in China, Int J Appl Earth Obs, 48, 28-36, doi:10.1016/j.jag.2015.09.009, 2016.

Dorigo, W., de Jeu, R., Chung, D., Parinussa, R., Liu, Y., Wagner, W., and Fernandez-Prieto, D.: Evaluating global trends (1988-2010) in harmonized multi-satellite surface soil moisture, Geophysical Research Letters, 39, doi:10.1029/2012gl052988, 2012.

Feng, H. and Zhang, M.: Global land moisture trends: drier in dry and wetter in wet over land, Sci Rep, 5, 18018, doi:10.1038/srep18018, 2015.

Gu, X. H., Li, J. F., Chen, Y. D., Kong, D. D., and Liu, J. Y.: Consistency and Discrepancy of Global Surface Soil Moisture Changes From Multiple Model-Based Data Sets Against Satellite Observations, Journal of Geophysical Research-Atmospheres, 124, 1474-1495, doi:10.1029/2018jd029304, 2019.

Hirschi, M., Stradiotti, P., Preimesberger, W., Dorigo, W., and Kidd, R.: Product Validation and Intercomparison Report (PVIR): Supporting Product version v08.1. Deliverable D4.1 Version 1, ESA Climate Change Initiative Plus - Soil Moisture, doi:10.5281/zenodo.8320930, 2023.

Li, X. W., Gao, X. Z., Wang, J. K., and Guoa, H. D.: Microwave soil moisture dynamics and response to climate change in Central Asia and Xinjiang Province, China, over the last 30 years, J Appl Remote Sens, 9, doi:10.1117/1.Jrs.9.096012, 2015.

Preimesberger, W., Scanlon, T., Su, C.-H., Gruber, A., and Dorigo, W.: Homogenization of Structural Breaks in the Global ESA CCI Soil Moisture Multisatellite Climate Data Record, IEEE Transactions on Geoscience and Remote Sensing, 59, 2845-2862, doi:10.1109/tgrs.2020.3012896, 2021.

Rahmani, A., Golian, S., and Brocca, L.: Multiyear monitoring of soil moisture over Iran through satellite and reanalysis soil moisture products, Int J Appl Earth Obs, 48, 85-95, doi:10.1016/j.jag.2015.06.009, 2016.

Wang, S. S., Mo, X. G., Liu, S. X., Lin, Z. H., and Hu, S.: Validation and trend analysis of ECV soil moisture data on cropland in North China Plain during 1981-2010, Int J Appl Earth Obs, 48, 110-121, doi:10.1016/j.jag.2015.10.010, 2016.

Zheng, X. M., Zhao, K., Ding, Y. L., Jiang, T., Zhang, S. Y., and Jin, M. J.: The spatiotemporal patterns of surface soil moisture in Northeast China based on remote sensing products, J Water Clim Change, 7, 708-720, doi:10.2166/wcc.2016.106, 2016.

**Replies to RC2**: Anonymous Referee #2, 28 Dec 2023

This study investigates the ability of surface and root-zone soil moisture from multiple reanalysis and remote-sensing products in representing drought events in recent 20 years globally, and compares their differences in describing various drought metrics. Overall, this paper provides a comprehensive reference for selecting datasets for drought study. But the structure and conclusions of this article are not clear enough for including too many datasets and drought events, so I suggest a major revision before publication. The main suggestions are as follows.

General comments:

The authors should be more familiar to Europe, and nearly half of the 18 selected events occurred over Europe. So why not just focus on the ability of multiple datasets in characterising seasonal drought events in Europe? In Figures 6－7 and 10, the drought metrics show remarkably discrepancies between seasonal and multi-year events. Thus I suggest the reconsideration of the clarification.

We thank the reviewer for the valuable feedback. Please find in the following our replies (in blue), together with pasted actual changes from the manuscript in selected cases (in order not to overload this document). Else, we refer to the respective sections of the revised manuscript.

Based on the comments of Reviewer #1, we decided to reframe the study and focus on the potential of long-term satellite observations and selected reanalysis products for characterising soil drying. This includes i) long-term negative changes in soil moisture, and ii) agricultural drought events. This change in the focus is also reflected in a change of the manuscript structure as well as its title, which now reads: "Potential of long-term satellite observations and reanalysis products for characterising soil drying: trends and drought events".

Thus, we first focus on the global soil moisture trends, which are based on the full year instead of dry season only (Section 4.1). Using the ACTIVE, PASSIVE and COMBINED satellite products, we identify regions with soil moisture trend direction agreement and those with trend disagreement (products deviate) in order to identify the areas where the agreement leads to higher confidence in satellite observed trends. We confront this with a similar analysis based on the reanalysis products.

In a new section "5.1 Synthesis on soil moisture trends", we then provide a synthesis of the global trends based on the "best-estimate" products from both remote sensing and reanalysis data. This synthesis is based on the analysis of the areas with trend direction agreement (cf. Fig. 9 of the revised manuscript) and considers the area fractions of positive and negative trends (cf. Table 2 of the revised manuscript).

We further investigate seasonal drought events as a use case to show how the diverse trend representation also affects the drought-detection capacity of the products (Section 4.3 – 4.5). For this, the product intercomparison of the drought metrics, particularly regarding the relation of product deviations in the drought magnitude and the soil moisture trends, is stratified by separating the drought regions in areas with drying trends agreement and in

those without trend direction agreement (Section 5.2). This stratified analysis based on the trend agreement allows to generalise the analysis of the impact of the trend representation on the products' drought-assessment capacity. We consider seasonal events only in the drought analysis (and neglect the original two multi-year events) in order to not overload the paper and to allow better comparability of the events.

Specific comments:

1.  The description of data and methods (section 2 and 3) are too long. Although the detailed information may be helpful to readers, it is not suitable in a scientific paper.

    We shortened the description of the datasets. In particular, we removed the C3S soil moisture product, since it is based on a precursor version of the processing algorithm of ESA CCI and thus does not represent the latest product achievements of merged satellite products. Also, as a suggestion from Reviewer #3, referenced literature on the validation of the products was moved and is now considered in the discussion (Section 5.1) to link the findings of our analysis to previous work.

2.  The figures and tables are not well organized in the paper structure. The quantitative results in tables can be integrated to the respective figures, which can make it more clear and comparable to readers. For example, the area mean of severity, magnitude and duration in Table 2 can be added to Figure 1－3, and the maximum of spatial extent of the events to Figure 5. In addition, Figure 4－5 can also be integrated in a Figure as (a) and (b), respectively.

    We thank the reviewer for the detailed suggestions on the organisation of the figures and tables. We integrated the numbers of the original Table 2 into the corresponding figures and adjusted the manuscript accordingly (new Figs. 3–5). We also combined the original Figures 4 and 5 into a two-panel figure as suggested (new Fig. 6).

3.  In term of the evaluation for the selected drought events, more statistical metrics can be included, such as pattern correlation, RMSE, and so on. Figures 6－9 are displayed only in bars, which is not concise and explicit enough. I recommend the Table graphic type to present each evaluation result for all events and all datasets. The detailed procedure can be seen at https://www.ncl.ucar.edu/Applications/table.shtml.

    Indeed, the presentation of the drought response as barplots is not ideal. We considered the proposed table graphic presentation of these results in the revised manuscript as new Fig. 7 (also included below).

[Figure]

Figure 7 Drought metrics of recent major drought events. The values are based on surface soil moisture and root-zone soil moisture (products denoted with *) and represent the area mean over the respective core of the event region in case of severity, magnitude, and duration, and the temporal maximum in case of the spatial extent. NA is displayed when products do not exhibit standardized anomalies below −1.5 for a specific event.

4. The analysis of dry-season soil moisture is less related with the research objective. I think it is more reasonable to further compare the soil moisture during drought events after presenting the results for multiple drought events.

   As indicated in the replies to Reviewer #1, the dry-season trends are no longer used, and we refocused the study on soil moisture trends based on the full year (but excluding the soil frost period).

5. As for the long-term trend, the analysis may be better to be conducted for the drought events rather than another indicator.

   We do not think that trends based on the events are meaningful in this case since the events are scattered in space and time. But as mentioned, we restructured the analysis and first focus on the global soil moisture trends.

The discussion section is not convincing and substantial. In 5.1, for drought metrics and dry-season SM trend were derived from the same variable, they must be related. In 5.2, the attribution method is too simple and no quantitative results are shown.

By reframing the study and focussing on the potential of long-term satellite observations for characterising soil drying, we investigate seasonal drought events as a use case to analyse the impact of the diverse trend representations on the drought-detection capacity of the products. For this, the analysis on the relation of product deviations in the drought magnitude and the soil moisture trends is stratified by separating the drought regions in areas with drying trends agreement and in those without trend direction agreement (Section 5.2, Fig. 10). Hence, the aim is to document the impact of uncertainties in the trend

representation on the trend-drought relation rather than point to its existence. This analysis shows that consensus in the soil moisture drying results in more consistent drought signals of the products and thus indeed highlights the importance of the trend representation on the drought-detection capacity of the products.

As for the original Section 5.2 (new Section 4.2 in the revised manuscript), we added statistical metrics (e.g., pattern correlations, mean bias and RMSD between the reanalyses and gridded observations, cf. new Supplementary Table 3, see below) to better attribute the differences in soil moisture trends to the driving variables. Also, this is accompanied by a discussion of studies that validate the considered products with reference soil moisture data to link our results to previous work (Section 5.1). The discussed literature supports our findings on the biases in MERRA-2 (cf. Section 4.2) but also ESA-CCI-ACT soil moisture trends.

**Supplementary Table 2 Validation of global patterns of precipitation and temperature trends. Metrics are based on the comparison to trends in CRU gridded observations (cf. Fig. 2).**

|  | Metric | ERA5 | ERA5-Land | MERRA-2 |
|---|---|---|---|---|
|  | Correlation | 0.33 | NA | 0.34 |
| **Precipitation** | Mean bias (mm/d yr$^{-1}$) | −0.0002 | NA | 0.0086 |
|  | RMSD (mm/d yr$^{-1}$) | 0.025 | NA | 0.042 |
|  | Correlation | 0.65 | 0.7 | 0.65 |
| **Temperature** | Mean bias (K yr$^{-1}$) | 0.011 | 0.012 | −0.011 |
|  | RMSD (K yr$^{-1}$) | 0.027 | 0.026 | 0.031 |

**Replies to RC3**: Anonymous Referee #3, 28 Dec 2023

The study investigates the ability of active and passive based remote sensing soil moisture products and land reanalyses to capture documented drought events and drought trends during the period 2000-2020. The drought events are characterised in different parts of the world by their severity, duration and spatial extent. The events are placed in the context of dry season soil moisture trends and potential reasons for diverging soil moisture trends between the different products are investigated. It is found that all the products capture the selected drought events. Significant differences between the products are found – for example, responses in surface soil moisture tend to be weakest for the active remote sensing products. For the global reanalyses, ERA5 and ERA5-land have a greater tendency for drying trends, whilst MERRA-2 has a greater tendency for wetting trends. Based on other reanalysis variables (evapotranspiration, runoff, precipitation) and observational data, it would appear that the ERA5 and ERA5-land trends are more reliable overall.

The authors have done a detailed and robust evaluation of the different products and have done well to disentangle the reasons (or potential reasons) for the divergences in the results. However, I think the introduction and discussion sections need to be more concise, with some of the detail removed. Further, I think the paper could be strengthened by linking the results to studies where reference soil moisture datasets (e.g. in situ data) have been used to validate drought events and trends (e.g. Li et al., 2020). This would give more weight to the conclusions of the study. Furthermore, I think the rationale for the approach used in this study needs to be more clearly communicated in the abstract and conclusion. Please also see the minor comments below.

We thank the reviewer for the positive feedback. Please find in the following our replies (in blue).

As a response to Reviewer #1, we decided to reframe the study and focus on the potential of long-term satellite observations and selected reanalysis products for characterising soil drying, which includes i) long-term negative changes in soil moisture, and ii) agricultural drought events. Thus, we first focus on the global soil moisture trends, and in a second step investigate the agricultural drought events as a use case to show the impact of the diverse trend representation on the drought-detection capacity of the products. This change in the framing of the study is also reflected in the introduction and the discussion sections, as well as in a change of the overall manuscript structure and title, which now reads: "Potential of long-term satellite observations and reanalysis products for characterising soil drying: trends and drought events".

To strengthen the link with existing literature, we moved the referenced literature on the validation of the products, which were cited in the dataset section, to the Discussion section and extended it. This also helped to shorten Section 2 as requested by Reviewer #2.

Line 46: Replace "trigger" with "triggers"

Was replaced.

Lines 76-94: I agree with the rationale for the evaluation approach. However, I still think the authors should link the results to studies where reference datasets have been used in the discussion section (5), as it would reinforce the findings in this study.

We now discuss several validation studies that evaluate the considered remote-sensing and reanalysis products with situ observations in the new section "5.1 Synthesis on soil moisture trends" to link our results to previous work. The discussed literature supports our findings on the biases in MERRA-2 (cf. Section 4.2) but also ESA-CCI-ACT soil moisture trends.

Line 148: Suggest to replace "as for" with "to"

The description of C3S soil moisture was removed since the product is not considered anymore in the revised manuscript (cf. reply to Reviewer #2).

Section 2.1.2 ERA5

It is important to mention here that in ERA5, T2m/RH2m pseudo observations are assimilated in the soil moisture analysis (see for example de Rosnay et al., 2013). These observations tend to have an important impact on root-zone soil moisture and latent/sensible heat fluxes with the atmosphere (see e.g. Fairbairn et al., 2019). The sensitivity of ERA5 to drought events could potentially be increased by assimilating these observations.

We thank the reviewer for this additional information. We now mention the assimilation of T2m/RH2m pseudo-observations in the ERA5 description and its effect on root-zone soil moisture and latent/sensible heat fluxes in the discussion.

layer 2 at 7–28 cm; layer 3 at 28–100 cm; and layer 4 at 100–289 cm. Apart from the assimilation of 2 m temperature and relative humidity pseudo-observations (e.g., De Rosnay et al., 2013), ERA5 is the first ECMWF reanalysis that includes remotely-sensed observations in a soil moisture analysis. Remote-sensing soil moisture from scatterometers (ERS-1,-2;

an assimilation of these ground data. Also, the assimilation of 2 m temperature and relative humidity in the soil moisture analysis of ERA5 tend to have an important impact on root-zone soil moisture and latent/sensible heat fluxes (Fairbairn et al., 2019), which could contribute to the increased sensitivity of ERA5 to drought events.

Line 381: Suggest to replace "the average" with "average"

Was replaced.

Line 399: Suggest to replace "of" with "for"

Was replaced.

Line 401-403: Please rephrase for clarity and maybe split into two sentences.

This sentence has been removed since these absolute values are difficult to interpret.

Line 497: Suggest to replace "and display" with "display"

Was replaced.

Line 526: Sentence starting with "Despite the considerable spread…" Please rephrase as sentence does not make sense.

The sentence has been removed.

Line 532: Suggest to replace "largest deviations" with "the largest deviations"

Was changed accordingly.

Line 564: Sentence starting with "These regional differences…". Please rephrase for clarity.

We split and rephrased this sentence.

The regional product differences in evapotranspiration and soil moisture trends also show a link to regional differences in 2 m temperature trends (Fig. 2 m–o). In the mentioned regions, the temperature trends for MERRA-2 are (more) negative, while ERA5/ERA5-Land show positive or only weak negative temperature trends. As for the precipitation trends, the

Line 570: Suggest to replace "of MERRA-2" with "for MERRA-2" and replace "of ERA5" with "for ERA5".

Changed accordingly.

References:

De Rosnay, P., Drusch, M., Vasiljevic, D., Balsamo, G., Albergel, C. and Isaksen, L., 2013. A simplified extended Kalman filter for the global operational soil moisture analysis at ECMWF. *Quarterly Journal of the Royal Meteorological Society*, *139*(674), pp.1199-1213.

Fairbairn, D., de Rosnay, P. and Browne, P.A., 2019. The new stand-alone surface analysis at ECMWF: Implications for land–atmosphere DA coupling. *Journal of Hydrometeorology*, *20*(10), pp.2023-2042.

---

## Referee Report (RR1)

This study investigates the ability of surface and root-zone soil moisture from multiple reanalysis and remote-sensing products in representing drought events in recent 20 years globally, and compares their differences in describing various drought metrics. Overall, this paper provides a comprehensive reference for selecting datasets for drought study. Although the authors have made a major revision in the whole storyline and figures, but I still suggest a major revision before publication. The main suggestions are as follows.

General comments:

1. Throughout the whole paper, the quantitative evaluation is still not sufficient, and there are too many qualitative statements, Such as Line 395, conclusions and abstract. For the multiple datasets used in the study, such reanalysis is clearly enough to readers.

2. Figure 7: It is better to show their differences with respect to the baseline dataset, and thus it is easier to capture their abilities. In addition, the statistical results, such as RMSE and patter correlation coefficients, can also be presented in this way.

3. Figure 8: I think it is more reasonable to intercompare the datasets for each drought events than all events.

4. Figure. 10: Except for the long-term trend, drought events are also largely affected by the interannual variability. Hence I suggest the authors add the relevant evaluation for the interannual variability.

Specific comments:

The numbers under all colorbars are too small, and it is better for the units of trend to transformed to *** (20yr)$^{-1}$

---

## Referee Report (RR2)

I thank the author for their efforts in addressing my comments, and I can see that the revised manuscript improves a lot. The aim, innovation, and implication are now clearer. However, I still have some comments as follows, before consideration of its publication.

1. The product deviations in drought magnitude showed a significant relation with deviations in the soil moisture trends in areas without trend direction agreement. How about other drought characteristics used in this study (e.g., spatial extent, drought severity, and frequency)? Some discussions are needed at least.

2. The units of label bar in Figure 1, 2, 9 should be revised. For example, "m3 m-3 year-1" should be changed to "$m^3 \, m^{-3} \, year^{-1}$"

3. Is it accurate to use "days*1" as the unit of drought severity? The severity is the accumulated time accumulated standardised anomalies over the whole drought period. It looks like that using "[1]" is more appropriate. For example, the units for accumulated precipitation deficit is "mm" instead of "mm*days".

4. Compared to precipitation (P), evapotranspiration (ET), and runoff (RNOF), the P-ET-RNOF is more related to the soil moisture anomaly. So I suggest to add the trend of P-ET-RNOF in the Figure 2.

5. The abstract is very long. Please make sure the length of abstract meets the requirement of HESS.

---

## Author Response (AR2)

**Anonymous Referee #1, 27 Jun 2024 (Report #2)**

I thank the author for their efforts in addressing my comments, and I can see that the revised manuscript improves a lot. The aim, innovation, and implication are now clearer. However, I still have some comments as follows, before consideration of its publication.

We thank the reviewer for the positive feedback and for recognising the improvements over the original version. Below are our responses to the remaining comments.

1. The product deviations in drought magnitude showed a significant relation with deviations in the soil moisture trends in areas without trend direction agreement. How about other drought characteristics used in this study (e.g., spatial extent, drought severity, and frequency)? Some discussions are needed at least.

We have included a corresponding figure for drought severity in the supplementary material (Supplementary Fig. 3, also included below) and added a statement on these results in the manuscript. Similar as for the drought magnitude, deviations in drought severity show a significant relation with deviations in the soil moisture trends only in areas with no agreement in trend direction.

[Figure]

Supplementary Figure 3 As Fig. 11 of the main manuscript, but for product deviations in drought severity as a function of product deviations in the 2000–2022 soil moisture trends.

2. The units of label bar in Figure 1, 2, 9 should be revised. For example, "m3 m-3 year-1" should be changed to "$m^3$ $m^{-3}$ $year^{-1}$".

All units are now denoted with exponents (e.g., "$m^3$ $m^{-3}$ $yr^{-1}$").

3. Is it accurate to use "days*1" as the unit of drought severity? The severity is the accumulated time accumulated standardised anomalies over the whole drought period. It looks like that using "[1]" is more appropriate. For example, the units for accumulated precipitation deficit is "mm" instead of "mm*days".

Indeed, this is more appropriate. We have changed the units of severity to "[1]".

4. Compared to precipitation (P), evapotranspiration (ET), and runoff (RNOF), the P-ET-RNOF is more related to the soil moisture anomaly. So I suggest to add the trend of P-ET-RNOF in the Figure 2.

We thank the referee for this suggestion. Please find below the trends based on the yearly means of cumulated monthly P-ET-R (calculated on annual basis). As can be seen, the trends in the annual terrestrial water balance (or terrestrial water storage) show a relation to the trends seen in soil moisture, but also some differences. ERA5 and ERA5-Land particularly show more widespread wetting in terrestrial water storage than in root-zone soil moisture, while MERRA-2 shows more widespread drying. These differences are due to the fact that components other than root-zone soil moisture (i.e., deeper layer soil moisture and groundwater, snow, ice, biomass water) also contribute to terrestrial water storage and its trends. In addition, during the data assimilation water may be added or removed in the soil moisture analysis of the reanalysis systems, leading to a non-closed water balance. This may explain the differences seen between ERA5 and ERA5-Land, as the former is directly affected by the data assimilation, while the latter is produced in offline mode.

We added these trends in P–ET–R in the supplementary material (Supplementary Fig. 2) in order not to overload Fig. 3 (original Fig. 2).

[Figure]

**Supplementary Figure 2 As Fig. 3 of the main manuscript, but for Theil-Sen trends on yearly means of the cumulated monthly terrestrial water balance (i.e., precipitation minus evapotranspiration minus runoff). The terrestrial water balance is cumulated on annual basis.**

differences in the soil moisture trends (Fig. 3 j–l, cf. Sect. 4.1). Supplementary Fig. 2 also shows the trends on yearly means of the cumulated monthly terrestrial water balance (i.e., precipitation minus evapotranspiration minus runoff, cumulated on annual basis). These trends in the annual terrestrial water balance (or terrestrial water storage) also show a relation to the trends seen in soil moisture, but also some differences. ERA5 and ERA5-Land particularly show more widespread wetting in terrestrial water storage than in soil moisture, while MERRA-2 shows more widespread drying. These differences are due to the fact that components other than root-zone soil moisture (i.e., deeper layer soil moisture and groundwater, snow, ice, biomass water) also contribute to terrestrial water storage and its trends.

5. The abstract is very long. Please make sure the length of abstract meets the requirement of HESS.

We shortened the abstract by removing unnecessary methodological details.

**Anonymous referee #2, 18 Jun 2024 (Report #1)**

This study investigates the ability of surface and root-zone soil moisture from multiple reanalysis and remote-sensing products in representing drought events in recent 20 years globally, and compares their differences in describing various drought metrics. Overall, this paper provides a comprehensive reference for selecting datasets for drought study. Although the authors have made a major revision in the whole storyline and figures, but I still suggest a major revision before publication. The main suggestions are as follows.

We thank the reviewer for the re-examination of our manuscript and the additional feedback. Below are our responses to the comments.

General comments:
1. Throughout the whole paper, the quantitative evaluation is still not sufficient, and there are too many qualitative statements, such as Line 395, conclusions and abstract. For the multiple datasets used in the study, such reanalysis is clearly enough to readers.

We extended the quantitative evaluation of the trends by adding a new Fig. 2 that builds upon and expands the original Table 2 with the global mean trends of the products as well as their mean trends in the wetting and drying areas respectively (see below; note that the original Table 2 has been moved to Supplementary Table 2). This additional analysis shows that not only the area fractions of the trend directions diverge between the products, but also their trend magnitudes. ESA-CCI-ACT and MERRA-2 show positive global means of the trends, while all other products show negative global means. The mean trend magnitudes in the wetting areas are largest for surface soil moisture of ESA-CCI-ACT and -PAS, as well as MERRA-2 (0.03–0.04 m$^3$ m$^{-3}$ (20 yr)$^{-1}$; cf. Supplementary Table 2). Both ESA-CCI-ACT and -PAS also show largest drying trend magnitudes (around -0.035 m$^3$ m$^{-3}$ (20 yr)$^{-1}$). The mean drying is somewhat lower, but largely consistent between the reanalysis products (around -0.03 m$^3$ m$^{-3}$ (20 yr)$^{-1}$) for both surface and root-zone soil moisture. The overall lowest trend magnitudes in both directions can be observed for ESA-CCI-COM and -RZSM (less than 0.02 m$^3$ m$^{-3}$ (20 yr)$^{-1}$).

In addition (see Point 2 below), we also extended the quantitative evaluation of the drought metrics by comparing the product deviations in the metrics with respect to the product median for the individual events, as well as by evaluating the products based on their spatial drought metrics patterns.

Regarding the statement in line 395, the quantitative numbers (bias, RMSD, correlation) are presented in the sections that proceed this concluding paragraph of Section 4.2.

[Figure]

**Figure 2 (a) Area fractions (in %) of wetting and drying trends within each product, as well as (b) their global mean trends and the respective mean wetting and drying trends (in m³ m⁻³ (20 yr)⁻¹). Note that trends are not masked for significance, but for common spatial coverage of the datasets. The values for the best-estimate products (cf. Sect. 5.1) are based on the areas with trend direction consensus. Note that the respective numbers that are referred to in the text can be found in Supplementary Table 2.**

Not only the area fractions of the trend directions diverge between the products, but also their trend magnitudes (Fig. 2 b). ESA-CCI-ACT and MERRA-2 show positive global means of the trends, while all other products show negative global means. The mean trend magnitudes in the wetting areas are largest for surface soil moisture of ESA-CCI-ACT and -PAS, as well as MERRA-2 (0.03–0.04 m³ m⁻³ (20 yr)⁻¹; Supplementary Table 2). Both ESA-CCI-ACT and -PAS also show largest drying trend magnitudes (around -0.035 m³ m⁻³ (20 yr)⁻¹). The mean drying is somewhat lower, but largely consistent between the reanalysis products (around -0.03 m³ m⁻³ (20 yr)⁻¹) for both surface and root-zone soil moisture. The overall lowest trend magnitudes in both directions can be observed for ESA-CCI-COM and -RZSM (less than 0.02 m³ m⁻³ (20 yr)⁻¹).

2. Figure 7: It is better to show their differences with respect to the baseline dataset, and thus it is easier to capture their abilities. In addition, the statistical results, such as RMSE and pattern correlation coefficients, can also be presented in this way.

We extended Fig. 8 (original Fig. 7; see below) with a table plot of the product deviations in the drought metrics with respect to the product median of each event as a baseline (cf. Fig. 8 b). The weaker drought representation of ESA-CCI-ACT becomes apparent and is particularly pronounced for events in Europe. Similarly, MERRA-2 root-zone soil moisture shows weaker droughts, in this case most evident for events in North America and Africa. The deviations for MERRA-2 surface soil moisture are more mixed, with a weaker representation of the East Africa 2015 drought, but a stronger representation (particularly in terms of duration and severity) of the Texas 2011 and the Iberian Peninsula 2011-2012 droughts. Also, ESA-CCI-COM-RZSM shows stronger drought representation for many events.

[revised manuscript text omitted]

3. Figure 8: I think it is more reasonable to intercompare the datasets for each drought events than all events.

Indeed, the intercomparison of the datasets for each drought event is shown in the extended Fig. 8 (original Fig. 7; see above), where the metrics for each event are displayed for all datasets. This is summarised for all events in this Fig. 9 (original Fig. 8).

4. Figure 10: Except for the long-term trend, drought events are also largely affected by the interannual variability. Hence, I suggest the authors add the relevant evaluation for the interannual variability.

We thank the reviewer for this suggestion. Based on this, we evaluated the relation of the product deviations in drought severity and magnitude to the product deviations in the inter-annual variability of the standardised soil moisture anomalies. The inter-annual variability is characterised by the standard deviation of the annual mean standardised soil moisture anomalies of the 2000–2022 period, which are detrended using a LOWESS filter. The new Supplementary Fig. 4 (included below) indeed indicates a significant relation between drought severity and the inter-annual variability of soil moisture (Supplementary Fig. 4 a). Thus, products with larger inter-annual variability in soil moisture display stronger drought severities. However, such a relation is not evident for the drought magnitudes, which show no significant relation with inter-annual soil moisture variability (Supplementary Fig. 4 b). This may be because magnitude represents only one day of each event (i.e., the temporal minimum of the standardised anomalies during the drought period), whereas severity is calculated as the accumulated standardised anomalies over all days below the drought threshold and thus tends to be more related to the annual mean of the anomalies. We added a paragraph on these additional analyses to Section 5.2.

[Figure]

**Supplementary Figure 4 Product deviations in (a) drought severity and (b) magnitude as a function of product deviations in the inter-annual variability of the standardised soil moisture anomalies. The inter-annual variability is characterised by the standard deviation of the annual mean standardised soil moisture anomalies of the 2000–2022 period, which are detrended using a LOWESS filter. Deviations are displayed with respect to the product median of the individual events, separately calculated for the surface and the root zone (the latter additionally indicated with a "+"), with circle sizes depending on the chronology of the events within the investigated period (i.e., later events are displayed with larger circles). The inter-annual variability and drought metrics are averaged over the respective drought regions. The p-values of the linear trend slope (dashed line) and the Spearman rank correlation rho between the drought metrics and the soil moisture trends are noted as well.**

Apart from the trend representation, also the inter-annual variability of soil moisture may contribute to the drought-detection capacity of the products. This is investigated by relating the product deviations in drought severity and magnitude to the product deviations in the inter-annual variability of the standardised soil moisture anomalies. The inter-annual variability is characterised by the standard deviation of the annual mean standardised soil moisture anomalies of the 2000–2022 period, which are detrended using a LOWESS filter. Supplementary Fig. 4 a indicates a significant relation between drought severity and the inter-annual variability of soil moisture. Thus, products with larger inter-annual variability in soil moisture display stronger drought severities. However, such a relation is not evident for the drought magnitudes, which show no significant relation with inter-annual soil moisture variability (Supplementary Fig. 4 b). This may be because magnitude represents only one day of each event (i.e., the temporal minimum of the standardised anomalies during the drought period), whereas severity is calculated as the accumulated standardised anomalies over all days below the drought threshold and thus tends to be more related to the annual mean of the anomalies.

Specific comments:
The numbers under all colorbars are too small, and it is better for the units of trend to transformed to *** $(20\text{yr})^{-1}$

We have increased the font sizes for the numbers and units under all colorbars. Also, we have scaled the trends to *** $(20\ \text{yr})^{-1}$ to enhance readability and interpretability of the numbers.